# CA-SSLR: Condition-Aware Self-Supervised Learning Representation for Generalized Speech Processing

**Yen-Ju Lu**[†]**, Jing Liu, Thomas Thebaud**[†]**, Laureano Moro-Velazquez**[†]**,**
**Ariya Rastrow**, **Najim Dehak**[†]**, Jesus Villalba**[†]
[†]Center for Language and Speech Processing, Johns Hopkins University
{ylu125, tthebau1, laureano, ndehak3, jvillal7}@jhu.edu

## Abstract

We introduce Condition-Aware Self-Supervised Learning Representation (CA-SSLR), a generalist conditioning model broadly applicable to various speech-processing tasks. Compared to standard fine-tuning methods that optimize for downstream models, CA-SSLR integrates language and speaker embeddings from earlier layers, making the SSL model aware of the current language and speaker context. This approach reduces the reliance on the input audio features while preserving the integrity of the base SSLR. CA-SSLR improves the model's capabilities and demonstrates its generality on unseen tasks with minimal task-specific tuning. Our method employs linear modulation to dynamically adjust internal representations, enabling fine-grained adaptability without significantly altering the original model behavior. Experiments show that CA-SSLR reduces the number of trainable parameters, mitigates overfitting, and excels in under-resourced and unseen tasks. Specifically, CA-SSLR achieves a 10% relative reduction in LID errors, a 37% improvement in ASR CER on the ML-SUPERB benchmark, and a 27% decrease in SV EER on VoxCeleb-1, demonstrating its effectiveness.

## 1 Introduction

The emergence of Self-Supervised Learning Representations (SSLRs) models has revolutionized speech processing, setting new standards in the field. Pioneering models like Wav2vec 2.0 Baevski et al. [2020], HuBERT [Hsu et al., 2021], and WavLM [Chen et al., 2022a] leverage unlabeled audio data to learn rich representations of spoken language. These models are pivotal in a wide range of applications, including Speech Recognition (ASR) [Chang et al., 2021], Speaker Verification (SV) [Chen et al., 2022b, Tak et al., 2022], Language Identification (LID) [Bartley et al., 2023], and Speech Translation (ST) [Tang et al., 2022]. Benchmarks such as SUPERB [Yang et al., 2021] and ML-SUPERB [Shi et al., 2023a] have been crucial in evaluating SSL model performance, providing standardized tasks.

Although SSLR training approaches combine speech from various sources, these models learn representations solely from unpaired audio-only data. When extending SSLR features to multilingual scenarios and low-resource languages, unsupervised training limits the model's ability to distinguish between different languages, resulting in unified features for all languages. Additionally, labeling all SSL training data with language and speaker information requires significant human effort and is impractical. Thus, a post-training conditioning approach is more favorable. In other fields, methods like [Zhang et al., 2023] and IP-Adaptor [Ye et al., 2023] in image processing, and CTRL [Keskar et al., 2019] in NLP, have successfully integrated conditioning into pretrained models, demonstrating potential applications for speech processing.

In response to these challenges, we propose Condition-Aware Self-Supervised Learning Representation (CA-SSLR), a generalist conditioning model applicable to various speech-processing tasks such

38th Conference on Neural Information Processing Systems (NeurIPS 2024).

as language identification, multilingual speech recognition, and speaker verification. Unlike standard adaptation methods that heavily revise the SSLR for downstream tasks, CA-SSLR minimally adjusts the pretrained model by integrating language and speaker embeddings from earlier layers, making the SSLR aware of the current language and speaker context. This technique enables the creation of models that perform multiple tasks with a single adapted SSL encoder by strategically injecting conditional adapters into each encoder block while keeping the pretrained encoder weights frozen. CA-SSLR employs a hierarchical self-adaptation structure in which adapters at each layer leverage intermediate task-specific embeddings derived from lower layers. Through attention mechanisms and linear modulation, CA-SSLR dynamically adjusts scaling and biasing, effectively tailoring the model's response to language and speaker contexts. Our initialization strategy enables the conditioning module to perform identity transformations, preserving the original model behavior when introducing new conditions. By applying channel-wise linear modulation and time-wise scaling without mixing channel dimensions, CA-SSLR avoids introducing new cross-channel couplings, thereby mitigating overfitting and catastrophic forgetting. We demonstrate its versatility and efficiency on three widely studied multilingual speech processing tasks—ASR, LID, and SV.

The key contribution of this work is a novel approach to condition SSLRs using limited supervised labels, resulting in generalized speech representations that maintain the underlying model's behavior while improving performance with minimal additional parameters. Specifically, CA-SSLR offers:

- **Hierarchical Dynamic Conditioning**: We develop attention-based conditional adapters that are integrated throughout the SSL model. By periodically leveraging language- and speaker-specific information extracted from previous layers, CA-SSLR dynamically tailors the model's behavior at each time step, offering robust adaptation to multilingual tasks.
- **Preservation of Pre-trained Weights with Efficient Parameter Utilization**: CA-SSLR leverages the pre-trained encoder by introducing lightweight, channel-wise scala $\gamma$ and bias $\beta$ adjustments. Since these adjustments do not mix information across different channels, the original encoder's behavior is minimally altered. This strategy ensures stable, parameter-efficient training while preserving the benefits of the pretrained model.
- **Harmonized Task Compatibility with Notable Performance Gains**: Our experiments show that CA-SSLR effectively reduces trainable parameters, mitigates overfitting, and excels in under-resourced and unseen tasks. Notably, CA-SSLR achieves a 27% relative reduction in LID error rates, a 37% improvement in ASR CER on the ML-SUPERB benchmark, and a 27% decrease in SV EER on VoxCeleb-1. These results highlight CA-SSLR's effectiveness in enhancing multilingual SSLRs while also minimizing computational overhead for multitask fine-tuning.

## 2 Related Work

**Self-supervised learning representation.** Self-Supervised Learning (SSL) models, such as Wav2Vec 2.0 [Baevski et al., 2020], HuBERT [Hsu et al., 2021], and WavLM [Chen et al., 2022a], leverage large-scale of unlabeled audio to learn expressive speech representations. These representations capture acoustic, phonetic, and semantic features, which can then be subsequently fine-tuned on smaller labeled datasets for tasks such as speech recognition [Yi et al., 2020, Zhao and Zhang, 2022], emotion recognition [Pepino et al., 2021], vocal intensity classification [Kodali et al., 2023], and speaker change detection [Zajíc and Kunešová, 2023]. In multilingual settings, Wav2Vec 2.0-XLSR [Babu et al., 2021] extends pre-training to diverse languages for cross-lingual transfer, while mHuBERT [Lee et al., 2021] expands HuBERT to effectively handle multiple languages. These multilingual extensions facilitate a broad range of tasks, including speech recognition [Chen et al., 2023a], spoken language understanding [Hu et al., 2024], and speech generation tasks [Li et al., 2024], across various languages. Meanwhile, the scope of evaluation has broadened from monolingual benchmarks like SUPERB [Yang et al., 2021] to multilingual ones such as ML-SUPERB [Shi et al., 2023a], reflecting growing demand for robust and scalable SSL representations.

**Adaptation Methods.** In many studies [Yang et al., 2021, Shi et al., 2023a, Chen et al., 2023a], the SSL representations (SSLR) remains frozen while a decoder is trained for a specific task. Because the same encoder is shared across tasks, this approach allows multiple tasks to be evaluated on a single speech signal with only one encoder run. However, such systems often underperform compared to approaches that adapt the SSLR to the target task—by fine-tuning the entire SSL encoder Chen

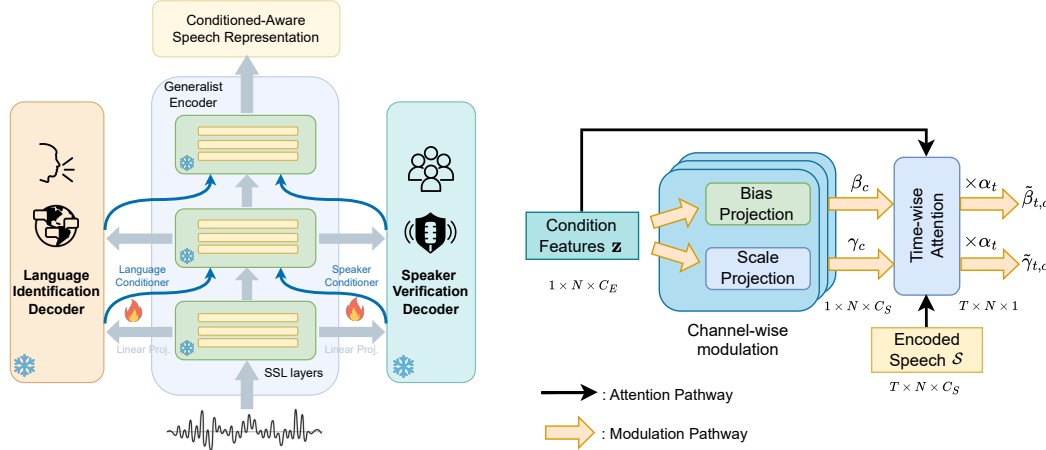

(a) CA-SSLR improves SSL features by integrating intermediate LID/SV conditions, keeping pretrained parameters frozen.

(b) The trainable time-channel attention conditioner for integrating language and speaker prediction in CA-SSLR. It predicts bias $\tilde{\beta}$ and scale $\tilde{\gamma}$ using condition feature $\mathbf{z}$.

Figure 1: CA-SSLR scheme and its time-channel attention conditioner. Only the conditioner and linear projections for the decoders are trainable, and all other parameters are frozen during adaptation.

et al. [2022a], fine-tuning only selected layers, or adding lightweight adapters [Chen et al., 2023b]. Unfortunately, these methods require a separate encoder for each task, leading to a significant increase in computational load that scales linearly with the number of tasks.

**Conditioning Pre-trained Models.** Image processing has successfully integrated conditioning into pretrained models using methods such as ControlNet [Zhang et al., 2023] and IP-Adaptor [Ye et al., 2023]. ControlNet enables precise control over generated images by incorporating additional inputs (e.g. edge maps or sketches), while IP-Adaptor utilizes lightweight adapter modules to modulate the model's behavior based on specific conditions—without modifying the pre-trained model's parameters. These techniques have achieved significant success and offer insights for potential applications in speech processing. Similarly, in Natural Language Processing (NLP), the Conditional Transformer Language Model (CTRL) [Keskar et al., 2019] introduces conditioning via control codes, guiding text generation based on attributes like style or domain and enabling efficient adaptation without extensive retraining. These successes in image processing and NLP demonstrate the potential for conditioning pre-trained SSLRs in speech applications.

**Hierarchical Conditioning.** Hierarchical modeling has also been explored in previous research. For instance, [Sanabria and Metze, 2018] propose a multi-task ASR model that applies intermediate representations by performing connectionist temporal classification (CTC) at multiple layers, each targeting different granularities. Lower layers predict character tokens, whereas higher layers predict subword units with increasingly larger vocabularies—scaling from 300 to 10k subword units in the final layer. [Chen et al., 2023a] extend this idea by incorporating hierarchical conditional layers into the ASR decoder, using tokens predicted by earlier layers to guide the predictions of subsequent layers.

## 3 Methodology

We introduce Condition-Aware SSLR (CA-SSLR), a universal encoder designed for multiple downstream speech tasks. CA-SSLR enriches a pre-trained SSL model by integrating intermediate LID and SV predictions to condition and adapt subsequent layers dynamically. This strategy allows the model to capture key language and speaker attributes, progressively refining outputs and excelling in multilingual, multispeaker scenarios. Figure 1a illustrates the CA-SSLR architecture. It comprises a frozen SSL encoder, which is augmented with trainable conditioners and lightweight task-specific decoders. The conditioner modulates the encoder's hidden representations based on conditioning features drawn from intermediate LID and SV embeddings. This hierarchical conditioning mechanism enables dynamic adaptation to a variety of input conditions while keeping the pre-trained

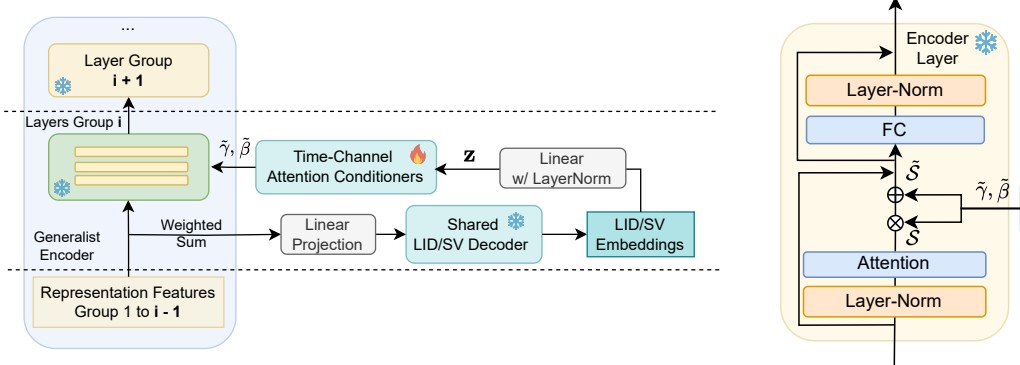

(a) Hierarchical conditioning with TCACs to generate feature **z** and modulate layers with scale $\tilde{\gamma}$ and bias $\tilde{\beta}$.

(b) SSLR layer with conditioning integration, transforming **S** into **S̃**.

Figure 2: Architecture of the CA-SSLR model employing hierarchical self-conditioning with Time-Channel Attention Conditioners (TCACs).

parameters fixed. The subsections below describe (1) the conditioner module, (2) how it integrates into the overall architecture, and (3) the incremental training strategy for incorporating conditioning information without catastrophic forgetting.

## 3.1 Channel-wise and Time-wise Attention Conditioner

A key component of CA-SSLR is the channel-wise conditioner (CC) or the time-channel attention conditioner (TCAC), which modulates the SSL encoder's hidden representations based on the conditioning features. As shown in Fig. 1b, the TCAC receives latent representations $\mathbf{S}^{(l)} \in \mathbb{R}^{C \times T}$ from layer $l$ of the SSL encoder and a conditioning feature vector $\mathbf{z} \in \mathbb{R}^{R}$, derived from intermediate LID or SV embeddings. The TCAC then produce modulated representations $\tilde{\mathbf{S}}^{(l)}$ by applying time- and channel-dependent scaling and bias:

$$\tilde{\mathbf{S}}_{t,c}^{(l)} = \text{TCAC}(\mathbf{S}_{t,c}^{(l)}, \mathbf{z}) = \tilde{\gamma}_{t,c}^{(l)}(\mathbf{z}, \mathbf{S}^{(l)})\mathbf{S}_{t,c}^{(l)} + \tilde{\beta}_{t,c}^{(l)}(\mathbf{z}, \mathbf{S}^{(l)}) \tag{1}$$

where $t$ and $c$ index time and channel dimensions, respectively. The scales $\tilde{\gamma}_{t,c}^{(l)}$ and biases $\tilde{\beta}_{t,c}^{(l)}$ are products of time- and channel-dependent components:

$$\tilde{\gamma}_{t,c}^{(l)}(\mathbf{z}, \mathbf{S}^{(l)}) = \alpha_t^{(l)}(\mathbf{z}, \mathbf{S}^{(l)}) \times \gamma_c^{(l)}(\mathbf{z}) \qquad \tilde{\beta}_{t,c}^{(l)}(\mathbf{z}, \mathbf{S}^{(l)}) = \alpha_t^{(l)}(\mathbf{z}, \mathbf{S}^{(l)}) \times \beta_c^{(l)}(\mathbf{z}) \tag{2}$$

The channel-dependent scales $\gamma^{(l)} \in \mathbb{R}^{C}$ and biases $\beta^{(l)} \in \mathbb{R}^{C}$ are computed by linear transformations of the conditioning feature, akin to feature-wise linear modulation [Perez et al., 2018]:

$$\gamma^{(l)}(\mathbf{z}) = \mathbf{W}_\gamma^{(l)}\mathbf{z} + \mathbf{b}_\gamma^{(l)} \quad \beta^{(l)}(\mathbf{z}) = \mathbf{W}_\beta^{(l)}\mathbf{z} + \mathbf{b}_\beta^{(l)} \quad \text{with },\tag{3}$$

while the time-dependent scales $\alpha^{(l)} \in \mathbb{R}^{T}$ are obtained via an additive attention mechanism:

$$\alpha_t^{(l)}(\mathbf{z}, \mathbf{S}^{(l)}) = \mathbf{v}_\alpha^{\text{T}} f(\mathbf{W}_\alpha^{(l)} \begin{bmatrix} \mathbf{S}_t^{(l)} \\ \mathbf{z} \end{bmatrix} + \mathbf{b}_\alpha^{(l)}) \tag{4}$$

where $f(\cdot)$ denotes a ReLU nonlinearity, $\mathbf{W}_\alpha^{(l)} \in \mathbb{R}^{C' \times (C+R)}$, $\mathbf{b}_\alpha^{(l)} \in \mathbb{R}^{C'}$, and $\mathbf{v}_\alpha \in \mathbb{R}^{C'}$. The conditioning feature $\mathbf{z}$ is produced by processing intermediate embeddings $\mathbf{e} \in \mathbb{R}^{E}$ from the LID or SV decoders, as $\mathbf{z} = \text{LayerNorm}(\mathbf{We} + \mathbf{b})$, where $\mathbf{W} \in \mathbb{R}^{R \times E}$ and $\mathbf{b} \in \mathbb{R}^{R}$ are shared linear parameters. If time-based modulation is not required, the simpler channel-wise conditioner (CC) can be used by retaining only the channel-dependent components $\gamma$ and $\beta$. This flexibility allows the model to be adapted for tasks with varying complexity requirements. By integrating these conditioning methods, CA-SSLR dynamically adapts its internal representations according to language and speaker features, without altering the frozen pre-trained encoder's parameters.

## 3.2 Hierarchical Self-Conditioning in CA-SSLR

CA-SSLR employs a hierarchical self-conditioning mechanism within the SSL encoder layers, building on the TCAC module. In Figure 2, the SSL encoder is segmented into layer groups, with TCACs inserted after each attention module to modulate hidden representations using updated conditioning features. The model aggregates SSL features through a weighted sum of outputs from all preceding layer groups, and this aggregate is passed to the LID and SV decoders. The decoders extract and transform LID and SV embeddings via a linear layer followed by layer normalization, forming the conditioning feature $\mathbf{z}$ used by the TCACs.

The conditioning feature $\mathbf{z}$ is re-estimated at intervals—every three layers for LID and every six layers for SV—using aggregated SSL features from earlier groups. This hierarchical design progressively refines the model's representations, adapting to the input's language and speaker characteristics at different depths of the network. For instance, the initial SSL layer group captures fundamental linguistic and speaker cues, producing embeddings that subsequently condition the next layer group via the TCAC modules. This ongoing refinement allows the model to adapt dynamically based on intermediate predictions, leading to context-aware, flexible representations.

Each layer group has distinct TCAC parameters, enabling fine-grained scaling and bias adjustments at different stages of the model. Notably, only the TCAC modules and the linear projections for the decoders are trainable, while all other SSL encoder parameters remain frozen throughout the conditioning process. This design mitigates overfitting risks and accelerates training by confining the number of learnable parameters. As a result, the hierarchical self-conditioning mechanism equips the model to capture diverse aspects of input audio, making CA-SSLR a robust framework for comprehensive speech analysis.

## 3.3 Incremental Training Strategy

Introducing new components into a pre-trained SSL encoder carries the risk of catastrophic forgetting. To address this, we use an incremental training strategy that gradually incorporates conditioning information. We initialize the TCAC parameters so that the initially modulated features match the original SSL outputs exactly. Specifically, we set $\alpha_t = 1$ for all $t$, $\gamma_c = 1$, and $\beta_c = 0$ for all $c$. According to Eq. (1), this ensures that $\tilde{\mathbf{S}}_{t,c}^{(l)} = \mathbf{S}_{t,c}^{(l)}$ at the outset, allowing a seamless transition from the pre-trained model to the conditioned model. For multiple conditioning features (e.g. LID and SV), each task's parameters $(\alpha_{\text{LID}}, \gamma_{\text{LID}}, \beta_{\text{LID}})$ and $(\alpha_{\text{SV}}, \gamma_{\text{SV}}, \beta_{\text{SV}})$ are initialized in the same manner (i.e., $\alpha = 1$, $\gamma = 1$, and $\beta = 0$). We then combine them as follows:

$$\alpha_{\text{total}} = \alpha_{\text{LID}} \times \alpha_{\text{SV}}, \quad \gamma_{\text{total}} = \gamma_{\text{LID}} \times \gamma_{\text{SV}}, \quad \beta_{\text{total}} = \beta_{\text{LID}} + \beta_{\text{SV}} . \tag{5}$$

By initializing each set of parameters to identity and then combining them, newly added conditioning tasks do not alter the features contributed by already integrated tasks. This allows the model to steadily integrate extra conditions without disrupting the knowledge acquired from previous tasks.

## 4 Experimental Setup

### 4.1 Datasets

We use the ML-SUPERB benchmark [Shi et al., 2023a] for both LID and ASR experiments. ML-SUPERB provides two data configurations: (i) 10-minute/language and (ii) 1-hour/languag, covering 123 well-represented languages (*Normal*). Both configurations also include five training utterances for each of 20 low-resource languages (*Few-shot*)[1]. For the *Few-shot* languages, we introduce an *Extended Few-shot* condition, which increases their data to match the *Normal* languages but provides only language labels without ASR transcripts. Our main goals are to address the severe LID constraints imposed by just five training utterances and to leverage the comparative ease of gathering LID-only data over fully transcribed data. For simplicity, we continue to refer to these extended data in subsequent LID experiments as *Few-shot*. As ML-SUPERB lacks speaker labels, we incorporate VoxCeleb2 [Nagrani et al., 2017] to train the models on the SV task. VoxCeleb2 contains 1,092 hours

---

[1]We found that some Lithuanian (lit) training and testing data are incorrectly replaced with Italian (it), so Lithuanian is excluded from the evaluation.

of speech from 5,994 speakers but lacks LID labels and ASR transcripts. SV performance is tested on the VoxCeleb1 original set. We augment the speech with Musan noise Snyder et al. [2015] and reverberation Ko et al. [2017] during SV training.

## 4.2 Model Architecture

**SSLR Models.** Our system leverages top multilingual SSL backbones from the ML-SUPERB benchmark: (i) Wav2Vec2-XLSR (300M parameters) [Babu et al., 2021], trained on 128 languages, and (ii) mHuBERT (100M parameter) [Lee et al., 2021], trained on English, Spanish, and French VoxPopuli [Wang et al., 2021] data. Both have demonstrated their efficacy in processing a wide range of linguistic inputs and form the backbone of our system. We conduct experiments using S3PRL [Yang et al., 2021] and ESPnet [Watanabe et al., 2018]. Our training set comprises data with (1) ASR+LID labels, (2) LID only labels, or (3) SV only labels; we computed the task-specific loss only when corresponding labels are available. Detailed information on the remaining hyperparameters is provided in the appendix[2]. Training a single model requires about one day on 2 A100 GPUs.

**Speaker and Language Decoders.** The speaker and language decoders are based on the ECAPA-TDNN architecture [Desplanques et al., 2020]. First, a convolutional layer projects the SSL representation to the decoder dimension (512 for LID, 1024 for SV). This is followed by one (for LID) or three (for SV) 1-dimensional SE-Res2Net [Gao et al., 2021] layers. Next, channel-wise attentive statistic pooling aggregates the frame-level features into an utterance-level vector, which is projected into lower-dimensional speaker embedding. We employ additive angular margin-softmax [Deng et al., 2019] with margin=0.3 for SV and margin=0.0 for LID. Large margin helps to create highly compact speaker representations [Villalba et al., 2022], while being detrimental in LID [Villalba et al., 2023]. Both the SV and LID decoders use a weighted average of all SSL layers for final predictions. In CA-SSLR, the intermediate embeddings that generate conditioning embeddings are similarly drawn from a weighted average of the SSL layers up to that point. Because all SV and LID decoders share parameters, the total number of trainable parameters does not increase with more frequent recomputation of conditioning embeddings.

**ASR decoder.** The ASR decoder conforms to the framework set by the ML-SUPERB benchmark [Shi et al., 2023a], facilitating comparable evaluations. A convolutional downsampling layer halves the SSL feature sequence duration. The resulting features are feed into a two-layer Transformer with 256-dims self-attention, eight heads, and a 1024-dims feed-forward layer. A linear output layer applies connectionist temporal classification (CTC) to predict multilingual character-level tokens.

# 5 Experiments and Results

## 5.1 Generalization Ability on Unseen Tasks

**Experiment Setting.** We conducted experiments to evaluate the generalization capabilities of the adapted SSLR models on LID, ASR, and SV tasks. The SSLR models are adapted for one task (either LID or ASR) and then evaluated on both the adapted task and an unseen task. For LID adaptation, the SSLR is trained exclusively with LID labels. We compared three setups: full fine-tuning (LID-FT), Houlsby adaptors Houlsby et al. [2019] (LID-Houlsby), and our proposed condition-aware approach (LID-CA-XLSR$_{\text{dual}}^{L}$). In this setup, we employed an additional LID decoder using the pre-trained SSLR to pre-generate language embeddings, which are then used to condition the SSLR model for a second inference pass. For ASR adaptation, the models are trained with ASR loss using three setups: full fine-tuning (ASR-FT), Houlsby adaptors (ASR-Houlsby), and our proposed hierarchical conditioning method with TCAC layers integrated into the SSLR model with single inference (ASR-CA-XLSR$^{L}$). During ASR adaptation, the LID decoder is integrated into the SSLR model to provide conditioning features, but SV information is not included during training.

**Results.** In LID adaptation (Table 1a), both LID-FT and LID-Houlsby improved LID performance compared to the pre-trained SSL baseline. However, on the unseen ASR task, the fully fine-tuned SSLR encoder improved ASR CER by only 2%, while LID-Houlsby showed limited generalization, with CER improvements of 5.4% and 3.9% for normal and few-shot languages, respectively. Our

---

[2]Code, pre-trained models, and reproduction instructions for the experiments are available at https://github.com/neillu23/CA-SSLR.

Table 1: Evaluation of adapted XLSR models on the 10-min ML-SUPERB and VoxCeleb dataset for LID, ASR, and SV tasks. These evaluations test the encoder's generalizability across different tasks, demonstrating effectiveness without further task-specific tuning.

(a) LID-adapted XLSR models evaluated on LID and ASR tasks.

| LID Adapted. | Bottleneck Dims. | LID Acc ↑ | | ASR CER ↓ | |
|---|---|---|---|---|---|
| | | Normal | Few-shots | Normal | Few-shots |
| XLSR | - | 89.1 | 83.9 | 29.0 | 39.0 |
| + LID-FT | - | 90.1 | 84.7 | 27.0 | 37.0 |
| + LID-Houlsby | 256 | 90.1 | 85.3 | 23.6 | 35.1 |
| + LID-CA-XLSR$_{\text{dual}}^{L}$ (ours) | 256 | **90.2** | **85.8** | **21.7** | **33.4** |

(b) ASR-adapted XLSR models evaluated on ASR and SV tasks.

| ASR Adapted. | Bottleneck Dims. | ASR CER ↓ | | SV | |
|---|---|---|---|---|---|
| | | Normal | Few-shots | EER ↓ | DCF ↓ |
| XLSR | - | 29.0 | 39.0 | 1.29 | 0.093 |
| + ASR-FT | - | **17.1** | 32.2 | 1.29 | 0.095 |
| + ASR-Houlsby | 256 | 20.3 | 34.6 | 1.37 | 0.097 |
| + ASR-CA-XLSR$^{L}$ (ours) | 256 | 18.6 | **31.6** | **1.15** | **0.088** |

LID-CA-XLSR$_{\text{dual}}^{L}$ method achieved substantially better generalization, improving ASR CER by 7.3% and 6.6% for normal and few-shot languages. In ASR adaptation (Table 1b), all models enhanced ASR performance, but ASR-Houlsby and full fine-tuning degraded SV performance relative to the baseline, highlighting their limited generalization. Our ASR-CA-XLSR$^{L}$ approach not only preserved but improved SV performance, reducing EER by relative 10.9% and DCF by 5.4%, showcasing strong generalization to the unseen SV task. These results demonstrate that CA-SSLR notably outperforms full fine-tuning and standard adaptation methods in terms of generalization. By effectively leveraging conditioning information, CA-SSLR adapts across tasks while maintaining performance on unseen ones. Our proposed conditioner offers both robust adaptations on training tasks and superior generalization, making CA-SSLR a versatile and effective solution for multilingual and multispeaker speech processing.

## 5.2 Condition-Aware SSLR Model

**Experiment Setting.** Table 2 investigates the CA-SSLR approach with hierarchical language conditioning. The first block of the table refers to the baseline where the foundational models are frozen, while the second block (CA-XLSR$_{\text{dual}}^{L}$) utilizes a separate task-specific LID model to pre-generate the language embedding. The third block presents our proposed approach, where we re-estimate the language embedding every fourth or third layer (CA-XLSR$^{L}$ (4L, 3L)) within the XLSR model, not required a separate LID system. The experiments utilized two types of conditioners: TCAC, which incorporates attention, and a variant without attention—referred to as channel-wise conditioners (CC)—where the same scale and bias are applied uniformly across all time frames. The real-time factors (RTF) as proc-time/signal-length are provided for assessing efficiency[3].

**Results.** First, we observed that both CA-XLSR$_{\text{dual}}^{L}$ and CA-XLSR$^{L}$ systems with TCAC (with attention) generally performed better than the CC (w/o attention) counterparts, reaffirming the benefits of the time-wise attention design. In the second block, CA-XLSR$_{\text{dual}}^{L}$ slightly outperformed CA-XLSR$^{L}$ in terms of CER for both the 10-minute and 1-hour datasets. However, its real-time factor (RTF) is akin to the combined RTFs of separate LID and ASR models since it runs Wav2Vec2 twice—once for language embedding extraction and again for ASR conditioning—posing challenges for streaming applications. On the other hand, CA-XLSR$^{L}$(CC, 3L) excelled among the three approaches, achieving a 35.9% and 19.0% relative improvement in Normal and few-shot languages,

---

[3]The RTFs are computed on an NVIDIA T4 GPU.

Table 2: ASR CER(%) and LID Acc (%) in ML-SUPERB 10min. and 1h. sets, comparing different layers to generate the language embedding to condition the following layers. We adapt the XLSR model for LID and ASR tasks.

| SSL Model | RTF↓ | REL. RTF↓ | 10mins | | | 1hr | | |
|---|---|---|---|---|---|---|---|---|
| | | | LID (ACC ↑) | ASR (CER ↓) | | LID (ACC ↑) | ASR (CER ↓) | |
| | | | Normal | Normal | Few-shots | Normal | Normal | Few-shots |
| XLSR [Shi et al., 2023a] | 0.021 | 1.00 | 66.9 | 29.2 | 40.9 | 87.9 | 22.0 | 39.3 |
| MMS-1b [Shi et al., 2023b] | - | - | 84.8 | 21.3 | 30.2 | 86.1 | 18.1 | 30.8 |
| XLSR (Ours) | 0.021 | 1.00 | 89.0 | 29.0 | 39.0 | 90.9 | 22.7 | 36.9 |
| CA-XLSR$_{dual}^{L}$(CC) | 0.037 | 1.75 | 89.0 | 18.6 | 32.2 | 90.9 | 14.1 | 31.5 |
| CA-XLSR$_{dual}^{L}$(TCAC) | 0.037 | 1.75 | 89.0 | **17.8** | 31.8 | 90.9 | **13.5** | 31.4 |
| CA-XLSR$^{L}$(CC, 4L) | **0.024** | **1.17** | **89.1** | 19.7 | 31.7 | 89.6 | 16.5 | 32.2 |
| CA-XLSR$^{L}$(CC, 3L) | **0.027** | **1.27** | 88.6 | 19.4 | **31.5** | 90.0 | 16.0 | 32.4 |
| CA-XLSR$^{L}$(TCAC, 3L) | **0.027** | **1.27** | 88.6 | 18.6 | 31.6 | **93.4** | 15.1 | **29.6** |

Table 3: Experiments on LID and LID + SV Hierarchical Conditioning. We adapt the XLSR and mHuBERT models for LID and ASR tasks using CA-SSLR$^{L}$, and for SV tasks using CA-SSLR$^{L,S}$. Results for Normal languages with 10-min and 1-hour datasets alongside VoxCeleb SV results.

| SSL Model | RTF↓ | REL. RTF↓ | 10min. ML-SUPERB + VoxCeleb | | | | 1h. ML-SUPERB + VoxCeleb | | | |
|---|---|---|---|---|---|---|---|---|---|---|
| | | | LID | ASR | SV | | LID | ASR | SV | |
| | | | ACC↑ | CER↓ | EER↓ | DCF↓ | ACC↑ | CER↓ | EER↓ | DCF↓ |
| mHuBERT | 0.015 | 1.00 | 81.9 | 38.2 | 2.19 | 0.145 | 86.2 | 30.9 | 2.19 | 0.145 |
| + FT | 0.015 | 1.00 | 73.0 | 36.5 | 5.85 | 0.350 | **87.7** | 32.3 | 4.01 | 0.251 |
| CA-mHuBERT$^{L}$(CC) | 0.017 | 1.13 | 82.0 | 31.9 | **1.77** | 0.120 | 86.1 | 25.1 | **1.77** | **0.118** |
| CA-mHuBERT$^{L,S}$(CC) | 0.018 | 1.16 | **82.2** | **31.7** | 1.79 | **0.117** | 87.3 | **24.8** | 1.78 | 0.121 |
| XLSR | 0.024 | 1.00 | 89.0 | 29.0 | 1.29 | 0.093 | 90.9 | 22.7 | 1.29 | 0.093 |
| + FT | 0.024 | 1.00 | 81.5 | 35.6 | 7.23 | 0.353 | 83.2 | 28.7 | 6.72 | 0.330 |
| CA-XLSR$^{L}$(CC) | 0.029 | 1.23 | 88.6 | 19.4 | 1.11 | 0.076 | 90.0 | 16.0 | 1.02 | 0.078 |
| CA-XLSR$^{L}$(TCAC) | 0.029 | 1.23 | 88.6 | 18.6 | 1.15 | 0.088 | 93.4 | 15.1 | 1.06 | 0.077 |
| CA-XLSR$^{L,S}$(CC) | 0.032 | 1.34 | **89.1** | 18.8 | **1.04** | **0.075** | 88.1 | 15.0 | **0.94** | **0.073** |
| CA-XLSR$^{L,S}$(TCAC) | 0.032 | 1.34 | 89.0 | **18.3** | 1.11 | 0.086 | **93.5** | **14.4** | 1.01 | 0.077 |

respectively, compared to the baseline in the 10-minute setup, and 33.5% and 19.8% in the 1-hour setup. LID accuracy remained comparable among the various CA-XLSR models, with a notable performance improvement from 90.9% to 93.4% in 1-hour setup.

## 5.3 Generalist Condition-Aware SSLR Model

**Experiment Setting.** Table 3 presents results for general CA-SSLR models that combine multilingual ASR, LID, and SV tasks. The table compares the baselines, with frozen and fine-tuned SSL models, to two different CA-SSLR Hierarchical models (CA-SSLR$^{L}$ and CA-SSLR$^{L,S}$). We further include another well-known multilingual SSLR model, mHuBERT, for a comprehensive comparison. The LID conditioning systems (CA-SSLR$^{L}$) are the same as from the previous section, conditioning the SSL model only on LID embeddings, with the SV decoder added on top of SSL features without further adaptation. The LID + SV conditioning system (CA-SSLR$^{L,S}$) combines both LID and SV embeddings and is jointly trained on ASR, SV, and LID losses. The intermediate LID embeddings are recomputed every three layers as the best configuration in Table 2, and SV embeddings are recomputed every six SSL layers. Apart from ASR CER and LID Acc on ML-SUPERB, we present SV equal error rates (EER) and detection cost function (DCF), measured at target prior probability $p = 0.05$ [Sadjadi et al., 2022], on VoxCeleb1. SV performance varied depending on whether we trained the model combining 10min ML-SUPERB + VoxCeleb2 or 1h ML-SUPERB + VoxCeleb2.

**Fine-tuning Baseline.** In the fully fine-tuning experiment, we initialized the model with pretrained ASR, LID, and SV decoders and fine-tuned for a few epochs. However, this approach resulted in suboptimal performance compared to the frozen SSLR baseline. The "FT" experiments showed degraded performance, with LID accuracy decreasing by 5.7%, ASR CER increasing by 3.1%, and SV EER worsening by 4.2 in absolute values on average across the four settings. This decline is unexpected, as fine-tuning typically improves performance. This suggests that simultaneous

Table 4: Ablation study of condition-aware settings for ASR-adapted XLSR models on 10-min ML-SUPERB dataset, using CC or TCAC. Conditioning is based on predicted language labels or LID embeddings, except in the ground truth (G.T.) experiment.

| ASR Adapted. | Normal CER ↓ | Few-shots CER ↓ |
|---|---|---|
| XLSR | 29.0 | 39.0 |
| + G.T. CC | 17.2 | 27.9 |
| + Hard CC | 19.8 | **28.6** |
| + Soft CC | 19.3 | 32.5 |
| + Embed CC | 18.6 | 32.2 |
| + Embed TCAC | **17.8** | 31.8 |

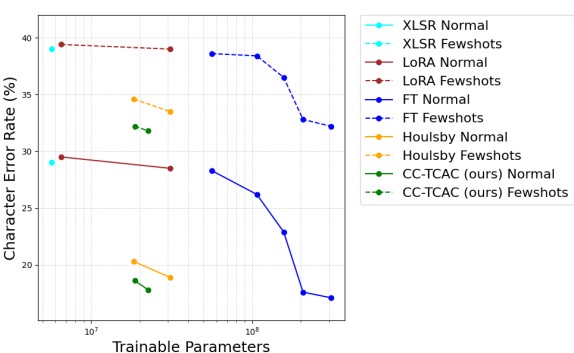

Figure 3: CER versus trainable parameters on XLSR model for Normal and Few-shots languages, demonstrating the adaptation ability for CC and TCAC.

adaptation of the SSL layers to multiple tasks causes conflicting adjustments, reducing the model's robustness. Consequently, catastrophic forgetting led to worse performance compared to the baseline. Conversely, the condition-aware SSLR models exhibited superior performance comparing with the frozen baseline, indicating that training the inserted condition layers does not alter the model's behavior for downstream tasks but improves its ability to represent the input speech data.

**Language Conditioned SSLR.** CA-SSLR$^L$(CC) notably enhanced SV performance w.r.t. the baseline, despite its encoder being solely tuned for ASR and LID tasks. For XLSR, the EER improved by 14% and 20% relative for the 10-min. and 1-h. sets, respectively, while DCF improved by 16-18%. Similarly, for mHuBERT, we observed comparable enhancements, with the EER improving by 17% in both sets and the DCF improving by 17-18%. This demonstrates that the CA-SSLR approach offers superior generalization capabilities compared to the original pre-trained SSL encoder, delivering improved performance. CA-SSLR$^L$(TCAC) performance in SV is comparable to its non-attention counterpart with better performance in LID and ASR as discussed in Sec.5.2.

**Language and Speaker Conditioned SSLR.** Adding a speaker conditioner to CA-SSLR$^{L,S}$ further improved its performance. In ASR tasks, incorporating the speaker conditioner to CA-XLSR$^{L,S}$ reduced CER by 3.1% for the 10-min. set and 6.2% for the 1-hr set, relative to CA-XLSR$^L$. For LID task, CA-SSLR$^{L,S}$ shows similar performance to other models with relative differences below 3%. For SV, CA-XLSR$^{L,S}$ using channel-wise conditioner (CC) reduced EER by 19.4-27.1%, outperforming CA-XLSR$^L$. Switching from CC to TCAC yielded additional gains in ASR, adding a relative improvement of 2.7-4.0%. In contrast, its impact on SV is more modest, with improvements in EER by 14.0-21.7%. Overall, TCAC demonstrated better adaptation ability, while CC excelled in generalization.

**ASR and RTF Discussion.** Generally, we observed the largest improvement for ASR when including the language conditioner, as it enables the system to adapt to produce output tokens in the correct language. Conversely, adapting the model to the input speaker provided fewer ASR gains. The XLSR model benefitted from our approach better than mHuBERT, possibly because mHuBERT is 3× smaller than XLSR, but more importantly, because mHuBERT is trained on just four languages compared to 128 in XLSR. Therefore, the pre-trained mHuBERT has not encountered enough diversity in terms of languages and speakers, thereby limiting its performance in multilingual ASR and SV. In terms of RTF, while the conditioned models are 13-34% slower compared to sharing the pre-trained SSL encoder for the three tasks, both CA-SSLR$^L$ and CA-SSLR$^{L,S}$ offer superior performance while being much faster than running task-specific models separately, indicating a more efficient use of computational resources while running the generalist model.

## 5.4 Analysis of the TCA Conditioner

**Ablation study of Conditioning Approach.** Table 4 conducts an ablation study for different conditioning methods with CA-XLSR$^L_{dual}$ settings within the ML-SUPERB 10min dataset regarding ASR CER. First, we used conditioners without attention (CC) on the ground truth LID predictions

(G.T.), serving as the upper bound for the performance of our proposed approach. This improved the *Normal* languages from 29.0% to 17.2%, and *Few-shot* languages from 39.0% to 27.9%, w.r.t. the pre-trained XLSR model. This showcases the potential of the condition-aware SSLR. Following, we compared conditioning on hard-predicted language labels (*Hard*), soft-predicted language labels (*Soft*), and language embeddings from the LID decoder bottleneck layer (*Embed*) for comparison. Conditioning on hard LID labels improved the most in *Few-shot* languages, improving by 26% relative to the baseline. On the other hand, the *Embed* case outperformed the *Soft* case and provided balanced performance for both *Normals* and *Few-shots* languages. Additionally, we compared CC to TCAC. The TCAC provided the best overall results, improving *Normals* and *Few-shots* by 38.6% and 18.5%, respectively, w.r.t. baseline.

**Parameter Efficiency in CER Reduction.** Figure 3 compares CER versus the number of trainable parameters for different adaptation methods, including our proposed Channel-wise Conditioner and Time-Channel Attention Conditioner (CC-TCAC), the Houlsby adapter, LoRA [Hu et al., 2021], full fine-tuning (FT), and the baseline XLSR model. The Houlsby adapters, with hidden dimensions of 256 and 512, have 18.4M and 30.9M trainable parameters. In comparison, the CC-TCAC approach, conditioned on precomputed LID embeddings with 256 dimensions (18.7M for CC and 22.6M for TCAC), achieves lower CERs with similar parameter counts. LoRA provided only marginal gains over the baseline, aligning with findings from Chen et al. [2023b]. In contrast, FT required fine-tuning 16–24 layers (200–300M parameters) to achieve comparable CER reductions, making CC-TCAC about ten times more efficient. As discussed in Sec 5.1, CC-TCAC's key contribution is its superior generalization ability. While the Houlsby adapter enhances task-specific adaptation, it falls short in generalizing to unseen tasks. In contrast, CC-TCAC achieves both effective adaptation and robust generalization, making it a versatile solution for diverse applications.

# 6 Conclusion

This paper introduces the CA-SSLR framework, an innovative approach that integrates conditioning into pre-trained Self-Supervised Learning (SSL) models by adapting only the trainable conditioner. Through a hierarchical self-conditioning mechanism, where intermediate language and speaker features condition the upper layers of the SSL model, CA-SSLR achieves a 33% reduction in ASR CER compared to the pre-trained baseline, matching the performance of single-task fully fine-tuned models. Additionally, it improves Speaker Verification EER by 27% and reduces Language Identification error rates by relative 10% in average. The results indicate that condition-aware SSLR models enhance the model's interpretation of input speech data, providing superior performance compared to traditional fine-tuning methods. This improvement is achieved by dynamically tailoring the model's response to the input language and speaker characteristics, ensuring robust generalization across various tasks. In summary, CA-SSLR offers a versatile and efficient approach to integrating conditioning information into pre-trained models. This method not only enhances performance across multiple tasks but also ensures efficient parameter utilization, supported by an improved RTF that facilitates its application in real-world scenarios.

**Broader Impact and Limitations** The CA-SSLR methodology improves the conditioning of pre-trained SSL models for speech processing, improving performance with minimal fine-tuning and reducing computational resource requirements. This advancement facilitates the deployment of robust models in resource-constrained environments, promoting broader access to advanced speech technology. Nevertheless, the methodology carries potential risks, as the conditioning mechanisms might amplify biases in the training data, leading to unfair outcomes, particularly for underrepresented languages and speaker groups. Ensuring diverse and balanced datasets, along with continuous monitoring, is crucial to mitigate biases and avoid reinforcing existing inequities.

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

# A  Model/Dataset Details and Training Hyper-parameters

This appendix provides detailed configurations and hyper-parameters for the decoder models used in our experiments, including ASR, LID, SV decoders, and the CA-SSLR models. The rationale behind specific hyper-parameter choices and architectural details are also discussed to offer insights into the experimental setup and model optimization strategies.

## A.1  Decoder Models for ASR, LID, and SV

The ASR, LID, and SV decoder models are optimized for their respective tasks through careful selection of hyper-parameters and architectural configurations. The ASR model directly follows the setting in ML-SUPERB benchmark [Shi et al., 2023a] for comparison. Table 5 summarizes these configurations. The "full" means one epoch is trained by passing all the training data.

Table 5: Hyper-parameters used for training ASR, LID, and SV decoder models.

|  | ASR | LID | SV |
| --- | --- | --- | --- |
| Feature Projection | 80 | 80 | 80 |
| Decoder Layers | 2 | 1 | 3 |
| Hidden Channels | 256 | 512 | 1024 |
| Dropout Rate | 0.1 | 0.3 | 0.0 |
| Loss | CTC | CE | CE |
| Learning Rate | 0.0001 | 0.0001 | 0.001 |
| Warmup Steps | - | - | 1000 |
| Effective Batch Size | 32 | 128 | 512 |
| Iterations per Epoch | 5000 | 5000 | full |
| Epochs | 20 | 10 | 20 |

Table 6: Training hyper-parameters for CA-SSLR models. The superscripts "Dec" and "Feat" represent the decoder and the feature projection layer, respectively.

|  | CA-SSLR$^{L}$ | | CA-SSLR$^{L,S}$ |
| --- | --- | --- | --- |
| Training Data | ML-SUPERB | VoxCeleb | ML-SUPERB + VoxCeleb |
| Condition Embedding | 256 (L) | | 256 (L) + 256 (S) |
| Condition Dropout Rate | 0.5 | | 0.5 |
| Initialization | ASR$^{\text{Dec}}$, LID$^{\text{Dec}}$, and SV$^{\text{Dec}}$ | | CA-SSLR$^{L}$ |
| Trainable modules | ASR$^{\text{Dec}}$, LID$^{\text{Feat}}$, and Adapters | SV$^{\text{Dec}}$ | ASR$^{\text{Dec}}$, LID$^{\text{Feat}}$, SV$^{\text{Feat}}$, and Adapters |
| Loss | ASR + LID | SV | ASR + LID + SV |
| Learning Rate | 0.0001 | 0.001 | 0.0001 |
| Effective Batch Size | 32 | 512 | 32 |
| Iterations per Epoch | 5000 | full | 5000 |
| Epochs | 20 | 20 | 20 |

## A.2  CA-SSLR Hierarchical Models

Table 6 provides detailed configurations for the CA-SSLR model. In the CA-SSLR$^{L,S}$ setup, two 256-dimensional embeddings are used to encapsulate language (L) and speaker (S) information, which then determine the parameters $(\alpha_{\text{L}}, \gamma_{\text{L}}, \beta_{\text{L}})$ and $(\alpha_{\text{S}}, \gamma_{\text{S}}, \beta_{\text{S}})$ following the procedure outlined in Eq. 5. The training adopts a stepwise approach, using initial parameters from an earlier phase to set up the next. The pretrained ASR, LID, and SV decoders serve as the foundation for initializing CA-SSLR$^{L}$; the SV decoder is fine-tuned further on top of CA-SSLR$^{L}$; and both CA-SSLR$^{L}$ and fine-tuned SV decoder initialize CA-SSLR$^{L,S}$. In the table's "Trainable modules" row, the notations LID$^{\text{Feat}}$ and SV$^{\text{Feat}}$ indicate that the feature projection layers of the LID and SV decoders are adjustable during

the training process. We conduct the experiments in Table 6 and Figure 3 multiple times, and the variations are all within 0.2% CERs range.

### A.3 Dataset License and Details

#### A.3.1 ML-SUPERB Dataset

The ML-SUPERB dataset is assembled from a wide collection of multilingual speech corpora, with each contributing corpus being governed by one of a variety of open-source licenses, such as Creative Commons, MIT, GNU, or Free-BSD. These licensing agreements guarantee that the dataset is openly available and can be used freely for both commercial and scholarly research purposes. The 10-minute training set encompasses 37.4 hours of data, and the 1-hour dataset increases the total to 222.4 hours of data. Additionally, the dataset includes development and testing sets, containing 41.8 hours and 45.0 hours of data, respectively. This dataset is designed for multilingual speech recognition and language identification, as used in our work.

In the original ML-SUPERB settings, there are two types of languages:

- Normal Languages: Each has 10 minutes or 1 hour of data per language, used for both LID and ASR training with transcriptions.

- Few-Shot Languages: Each has only 5 utterances. In the original settings, these languages are not presented in the results for LID training and are used for ASR training with available transcriptions.

For the extended few-shot condition, we incorporate the language labels from these few-shot data for LID training but continue using only 5 utterances with transcriptions for ASR training. Since language labels are more accessible than transcriptions, especially in low-resource scenarios. Table 7 summarizes the data configurations for the original and extended few-shot conditions.

Table 7: Data configurations for the original and extended few-shot conditions in ML-SUPERB.

| Data Per Language | Language Type | LID Training | ASR Training |
|---|---|---|---|
| Original Settings | Normal | 10 min – 1 hr | 10 min – 1 hr |
| | Few-Shot | Not presented in result | 5 utts |
| Extended Few-Shot | Few-Shot | 10 min – 1 hr (language labels only) | 5 utts |

#### A.3.2 VoxCeleb Dataset

The VoxCeleb dataset is available under the Creative Commons Attribution 4.0 International license and encompasses comprehensive training, development, and testing data collection. Specifically, it contains 1092 hours of audio from 5,994 speakers for training, 110 hours from 4,933 speakers for development, and 20 hours from 40 speakers designated for testing. Designed to facilitate speaker verification and identification tasks, aligns with our usage in the speaker verification task. To ensure privacy, speaker names within the dataset are anonymized and represented through unique speaker IDs.

## B  CER vs. Trainable Parameters

Table 8 compares the mHuBERT model's ASR performance against the number of trainable parameters, where the XLSR counterpart is shown in Fig. 3. Both CA-mHubert$_{\text{dual}}^{L}$ and CA-mHubert$_{\text{dual}}^{L,S}$ are with 256 condition feature dimensions. Notably, the CA-mHubert$_{\text{dual}}^{L}$ model excels in few-shots scenarios, while the CA-mHubert$_{\text{dual}}^{L,S}$ yields CERs for normal languages comparable to a fully fine-tuned 12-layer mHuBERT model using only 15.9M parameters. This efficiency demonstrates the TCAC's capability in the CA-SSLR framework to deliver fine-tuned levels of ASR accuracy with notably reduced parameter count, providing an optimal balance for practical ASR applications.

Table 8: Comparison of trainable parameters and CERs on ML-SUPERB 10min dataset, including fine-tuning top layers, LoRA, and dual-inference condition aware mHuBERT model.

| Approach | Trainable Params. | Normal CER ↓ | Few-shots CER ↓ |
|---|---|---|---|
| mHuBERT | 5.7M | 38.2 | 42.9 |
| LoRA | 15.2M | 38.3 | 42.7 |
| FT (2L) | 19.9M | 36.0 | 42.3 |
| FT (4L) | 34.0M | 35.0 | 42.3 |
| FT (6L) | 48.2M | 33.1 | 41.4 |
| FT (8L) | 62.6M | 32.0 | 40.5 |
| FT (12L) | 90.8M | **30.8** | 40.5 |
| CA-mHubert$_{\text{dual}}^{L}$ | 10.8M | 31.6 | **40.0** |
| CA-mHubert$_{\text{dual}}^{L,S}$ | 15.9M | **30.9** | 40.4 |

## C Training Efficiency and Resource Usage

We compare the training speed and resource consumption of different adaptation methods, including Houlsby Adapters, CA-SSLR, and full fine-tuning (FT). Table 9 summarizes the bottleneck dimensions, training times, and peak memory usage for each method. We evaluate the training speech for 10k iterations with batch size 8. We found that the CA-SSLR approach ranks second compared to the Houlsby Adapter and a fully fine-tuning approach in speed and memory usage. However, it is important to note that CA-SSLR surpasses the Houlsby Adapter in adaptation effectiveness and generalization ability, as demonstrated in Table 1. These results indicate that although CA-SSLR incurs a moderate increase in training resources, it provides benefits in performance and generalization. We acknowledge that the current implementation of CA-SSLR is not yet optimized for speed and memory efficiency. Future work will focus on optimizing the model to reduce training time and memory consumption without compromising performance.

Table 9: Comparison of adaptation methods in terms of bottleneck dimensions, training speed, and peak memory usage.

| Method | Bottleneck Dims. | Training Speed | Peak Memory Usage |
|---|---|---|---|
| Houlsby Adapter | 256 | 76 mins | 58 GB |
| CA-SSLR$^{L}$ (3L) | 256 | 120 mins | 68 GB |
| Fine-Tuning (FT) | - | 135 mins | 79 GB |

## D RTF Analysis

Tables 10 and 11 present the real-time factor (RTF) for each individual component, as well as for the combined systems. In Table 11, the term "separated tasks" refers to duplicating and fine-tuning the SSLR for each task individually, along with the corresponding total RTF.

## E Few-shots Results

Within the ML-SUPERB dataset's 20 few-shots languages, we examine the performance of CA-SSLR against the established SSL baselines, XLSR and mHuBERT, on models trained in the 10-minute ML-SUPERB set. The LID results indicate a close match between CA-SSLR and the baseline, with approximately half of the few-shot languages exhibiting improvements or matching their baseline performance. Section 5.4 reveals that SSL-based LID models are inherently effective, and extending full fine-tuning does not necessarily enhance results. This observation aligns with the outcomes of other classification tasks adeptly handled by SSL models, as documented in [Chen et al., 2023b]. Furthermore, the CA-SSLR framework demonstrates subtle enhancements for the Normal languages in the 10-minute set in Table 3, indicating that the LID performance remains robust despite the encoder's additional modifications.

| Modules | RTF |
|---|---|
| ASR Decoder | 0.004 |
| LID Decoder | 0.001 |
| SPK Decoder | 0.003 |
| XLSR SSL | 0.016 |
| CA-XLSR$^S$ (6L) | + 0.003 |
| CA-XLSR$^L$ (4L) | + 0.004 |
| CA-XLSR$^L$ (3L) | + 0.006 |
| mHubert SSL | 0.007 |
| CA-mHubert$^S$ (6L) | + 0.001 |
| CA-mHubert$^L$ (3L) | + 0.002 |

Table 10: RTF for different components.

| XLSR Approaches | RTF | mHubert Approaches | RTF |
|---|---|---|---|
| ASR + LID (Table 2) | | | |
| XLSR + ASR + LID | 0.021 | mHubert + ASR + LID | 0.013 |
| CA-XLSR$^L$(4L) | 0.024 | - | |
| CA-XLSR$^L$(3L) | 0.027 | CA-mHubert$^L$(3L) | 0.015 |
| Separated 2 tasks (+XLSR) | 0.037 | Separated 2 tasks (+mHubert) | 0.020 |
| ASR + LID + SV (Table 3) | | | |
| XLSR + ASR + LID + SV | 0.024 | mHubert + ASR + LID + SV | 0.015 |
| CA-XLSR$^L$ | 0.029 | CA-mHubert$^L$ | 0.017 |
| CA-XLSR$^{L,S}$ | 0.032 | CA-mHubert$^{L,S}$ | 0.018 |
| Separated 3 tasks (+2*XLSR) | 0.055 | Separated 3 tasks (+ 2*mHubert) | 0.030 |

Table 11: Total RTFs for combined systems.

Table 12: Evaluation of LID and ASR performance in terms of Accuracy (Acc) and Character Error Rates (CERs) for few-shot learning in low-resource languages using the ML-SUPERB 10-minute set, comparing on XLSR and mHuBERT models.

| Lang. | XLSR | | CA-XLSR$^{L,S}$ | | mHuBERT | | CA-mHuBERT$^{L,S}$ | |
|---|---|---|---|---|---|---|---|---|
| | Acc ↑ | CER ↓ | Acc ↑ | CER ↓ | Acc ↑ | CER ↓ | Acc ↑ | CER ↓ |
| bos | **82.0** | 21.3 | 70.0 | **11.7** | **30.0** | 29.0 | 28.0 | **26.0** |
| ceb | 97.6 | 20.5 | **97.6** | **12.4** | 92.9 | 27.2 | **97.6** | **25.4** |
| dan | **89.5** | 44.7 | 76.5 | **37.0** | 80.1 | 49.1 | **80.4** | **47.5** |
| epo | **81.7** | 15.3 | 76.9 | **14.5** | 46.2 | **24.8** | **52.9** | 25.4 |
| frr | 87.5 | 33.6 | **89.3** | **29.9** | **67.9** | 40.4 | 63.4 | **37.7** |
| ful | 55.0 | **27.1** | **67.5** | 28.2 | **37.5** | 34.7 | 62.5 | **32.0** |
| kaz | 98.0 | 32.6 | **99.3** | **21.5** | **91.4** | 37.8 | 88.7 | **36.0** |
| kea | 84.1 | 28.9 | **90.9** | 27.6 | 65.9 | 35.2 | **75.0** | **33.1** |
| lit | 87.3 | 52.0 | **87.7** | **45.2** | 79.3 | 52.4 | **82.5** | **49.3** |
| luo | **100.0** | 29.4 | 95.1 | **24.4** | 92.7 | **29.4** | 92.7 | 30.1 |
| srp | **64.8** | 57.4 | 50.9 | **48.1** | **53.5** | 56.7 | 45.7 | **56.2** |
| sun | 93.5 | 26.4 | **94.4** | **19.1** | **94.4** | 32.7 | 93.5 | **26.2** |
| tok | 98.5 | 15.4 | **98.5** | **13.1** | 98.5 | 23.2 | **98.5** | **19.2** |
| tos | **100.0** | 49.1 | 99.4 | **44.7** | 99.4 | 53.1 | **99.4** | **48.9** |
| tso | **84.0** | 25.4 | 81.3 | **21.7** | **83.3** | 29.4 | 81.3 | **26.0** |
| tsn | **87.1** | 22.7 | 85.7 | **17.3** | 83.6 | 27.3 | **84.3** | **23.9** |
| tur | **82.4** | 60.0 | 79.1 | **37.0** | **62.6** | 65.1 | 57.7 | **61.7** |
| umb | **64.0** | 24.6 | 40.0 | **23.3** | **44.0** | **29.8** | 36.1 | 30.1 |
| vie | **94.7** | 88.4 | 92.9 | **83.1** | **85.8** | **80.1** | 74.2 | 80.3 |
| zul | 60.6 | 20.4 | **62.3** | **14.4** | **53.7** | 24.3 | 52.0 | **20.4** |

Regarding the ASR results, most languages achieve significant CER reductions, ranging from a modest few percent to over 30%, when compared with SSL baselines. Notably, the Bosnian (bos) language experiences an impressive 45.1% relative improvement in CER, while Cebuano (ceb) improved by 39.5% with the XLSR model. With the mHuBERT model, the most substantial gains are observed in Sundanese (sun) and Toki Pona (took), with 19.9% and 17.2% CER relative improvements, respectively. These results underscore the CA-SSLR framework's profound effect in bolstering ASR performance for few-shot languages, especially demonstrating more pronounced improvements with the XLSR model.

When examining the correlation between LID accuracy and ASR performance, it is apparent that a lower CER does not necessarily align with high LID accuracy. For instance, Serbian (srp) on the XLSR model, despite having a modest LID accuracy of 50.9%, shows a CER improvement from 57.4% to 48.1%. Conversely, Fulah (ful), the sole language to exhibit a CER increase in the XLSR model, presents a higher LID accuracy of 67.5%. This indicates that the CA-SSLR framework's efficacy is not solely contingent on high LID prediction accuracy. CA-SSLR's reliance on embeddings instead of one-hot hard labels for predictions enables the model to maintain or improve ASR performance despite suboptimal LID scores. This approach allows the model to utilize embeddings to distinguish between easily confused languages, enabling the ASR model to predict the correct language accurately.

## F    Decode Examples

Table 13 visualizes ASR outcomes for the XLSR and CA-XLSR$^{L,S}$ models on the ML-SUPERB 10-minute dataset, covering both few-shot and standard language scenarios. It highlights CA-SSLR's superior language recognition capabilities and success in rectifying the misclassifications encountered with XLSR, often resulting in completely incorrect transcriptions. This is evident in languages such as Lithuanian and Turkish, categorized as *Few-shot*, and Bulgarian, which is better resourced (*Normal*). These findings demonstrate the TCA conditioner's effectiveness in accurately managing LID embedding features and distinguishing between languages for downstream tasks. Moreover, the results from other samples suggest that CA-SSLR achieves better outcomes during training due to its incorporation of language information, even when the XLSR model correctly predicts the language. This underscores the efficacy of the TCA conditioner in exploiting language-specific data, thereby enabling CA-SSLR to achieve heightened accuracy across a diverse range of languages.

## G    Ethical Statement

We affirm our commitment to ethical research practices, including respect for privacy and the responsible use of data. The proposed CA-SSLR model improves multilingual ASR in 143 languages, including 20 low-resource ones with just five training utterances—thus broadening access to speech technology for previously underserved communities. Furthermore, the model's focus on reducing computational costs at inference time helps lower both financial barriers and the environmental footprint of large-scale speech applications. Our experiments rely on publicly available datasets (ML-SUPERB and VoxCeleb), chosen for their moderate size to limit computational demands. We also employ widely used pre-trained models (XLSR and mHuBERT) in line with their intended research purposes, thereby adhering to community standards for reproducible and transparent research. Nevertheless, the capability to perform speech and speaker recognition raises concerns over potential misuse, such as the covert monitoring of private conversations by unauthorized entities or oppressive regimes intent on identifying dissidents. We therefore underscore the importance of public awareness and transparent dialogue about the implications of automated speech analysis, and the need for responsible governance of emerging technologies.

| Language | Group | Ground Truth | XLSR | CA-XLSR$^{L,S}$ |
|---|---|---|---|---|
| Esperanto (epo) | Few-shots | LI STUDVOJAĜIS AL ITALIO HIS-PANIO KAJ FRAN-CIO | LESSTOS VOLJAGIS AL LITALIO HIS-PANIO CKAI FRANCIO | LI STUD VOJAGIS AL ITALIO HIS-PANIO KAJ FRANZCIO |
| Lithuanian (lit) | Few-shots | KARALIUS NEIŠDRĮSO KALD-INTI VARINIŲ MONETŲ KAR-ALIŠKOJOJE MONETŲ KALYK-LOJE | КАРАЛЮОС НЕ ЖДРИСА КАЛЬНЕН ТЕ ВОАРИНУ МОНЕТУ КАРАЛЮШКО JОАЕ Б МОНАТУ КАЛІКЛОJА | KARALJUS NE IŽDR I SO KALNEN TIE VORINIU MONETU KARALJUŠKOJO JE MONETU KALIKLOJE |
| Serbian (srp) | Few-shots | OVO OTKRIĆE TAKOĐE PRUŽA UVID U EVOLU-CIJU PERA KOD PTICA | OVO ODKRIJČIE TAKO ĆE PRUŽA UVID UJEVOLUC I JU PERA KOB PTICT | OVO OD KRIČĆE TAKO ĐE PRUŽA UVID U EVOLUCI JU PERA KOD PTICT |
| Northern Frisian (frr) | Few-shots | MEI IK TAKOM WIKE DAT BOEK FAN DY LIENE | MEA EK TAKGKOMME WIGGE DAT BOEK VAN DIE IE LIEËNE | MAA IK TAKOMME WIKE DAT BOEK VAN IE LIENE |
| Turkish (tur) | Few-shots | TÜM BUNLAR ILGIMIZI ÇEKSE DE UZUN KALA-MAZDIK | ТYM БУНЛАР ИЛГИМЗЕЙ ЧАК СЕ ДЕ УЗУН КАЛАМАСДЫК | TÜM BUNLAR İLGİMİZİ ÇəKSEə DE UZUN KALA-MAZDIK |
| Belarusian (bel) | Normal | НА ПЕРШЫМ ПЛАНЕ КАРЦІНЫ НАМАЛЯВАНЫ ГОСЦІ НАКРЫТЫЯ ПЯЛЁСТКАМІ РУЖ | НА ПЕРШИМ ПЛАНЕ КАРЦИНЕ НАМАЛЕВАНЫЙ ГОСТЕ НА _КРЫТЫЕ ПЕЛЁСТКЫМІЕ РУЖ | НА ПЕРШЫМ ПЛАНЕ КАРЦІНЫ НАМАЛЯВАНЫЙ ГОСЦІ НАКРЫТЫЯ ПЕЛЁСТКАМІ РУЖ |
| Bulgarian (bul) | Normal | СЛЕД МАЛКО ВАСИЛЕНА ПАК ИЗЛЕЗЕ | SLED MAUKU VASILENA PAKI ИZLEZE | СЛЕД МАЛКО ВАСЕЛЕНА ПАК ИЗЛЕЗЕ |
| Basque (eus) | Normal | KORRONTE ETIKO HORREN HELBURUA ZO-RIONTASUNA LORTZEA DA | KORRONTE ETIKO HORREN HELBURUA ZO-RIOANTASUNA LORTZEA DA | KORRONTE ETIKO HORREN HELBURUA ZO-RIONTASUNA LORTZEA DA |
| Ndebele (nbl) | Normal | UMNQOPHO WOMSEBENZI LO | UMNCOPHO OWOMSEVENDZI LOU | UMNCOPHO WOMSEBENZI LO |

Table 13: The ground truth, predictions from XLSR and CA-SSLR models. Deletions are shown with red strikethrough text, insertions are underlined in blue, and substitutions are marked with yellow highlighting.

