# OpenReview forum: "CA-SSLR: Condition-Aware Self-Supervised Learning Representation for Generalized Speech Processing"
_NeurIPS.cc/2024/Conference — NeurIPS 2024 poster_

### Official Review · Reviewer_xgMo · 2024-06-18

**Soundness:** 2
**Presentation:** 1
**Contribution:** 3
**Rating:** 4
**Confidence:** 3

**Summary:**

This work proposes a method to automatically condition a (frozen) speech foundation model to a particular language and/or speaker. This method consists of 2 parts; they use an ECAPA-TDDN model to compute speaker or language embeddings from a set interval of intermediate layers of the SSL model, and a conditioning layer which uses those embeddings to modify subsequent layer outputs of the SSL model. The authors conduct experiments with the multilingual SUPERB benchmark (for language identification and ASR) and VoxCeleb1 dataset (for speaker recognition).


Edit Rebuttal: Houlsby adapter was addressed, presentation style was addressed but cannot be judged. Updated score from reject to borderline reject.

**Strengths:**

This work tackles the difficult problem of using a single model to perform ASR on 123 languages, with only 1 hour of training data for each language. They show that, during inference on the test set, making use of a ground-truth language label significantly improves the character error-rate, indicating that baseline models like wav2vec2-xlsr do not adequately activate the representations of a specific language. The authors propose a method where a model can learn to condition itself during inference on a specific target domain.

**Weaknesses:**

The presentation of this paper is, in my opinion, subpar to what is expected for NeurIPS. Most importantly, I found it difficult to grasp the proposed method from the methodology section. Line 123-131 are copied from the related work section. Line 133 introduces a “hierarchy” of conditioners, but hierarchy is never explained. Then line 140 assumes the reader is very familiar with TCAC while it has not been explained yet. Line 141 to 146 mention a lot of different configurations related to SV and LID decoders, this could be streamlined into a single “standard” setup, while the experimental section could ablate on different settings. Moreover, I think that the speaker decoder story only distracts from the interesting results of this paper (e.g., Table 1, Table 2, Figure 2 do not make use of the speaker decoder at all, so why make the method section hard to understand by including it?). Furthermore, I cannot implement the conditioner (TCAC) from the description in line 156-170. Also, line 221-222 should be in the method section.

In the experiment section, references to the appendix should be made to guide the reader to relevant information regarding experimental details. I cannot judge whether the current experimental setup is fair.

Furthermore, I think some relevant literature and methodology is missing:

[1] Houlsby, Neil, et al. "Parameter-efficient transfer learning for NLP." International conference on machine learning. PMLR, 2019.
[2] Thomas, Bethan, Samuel Kessler, and Salah Karout. "Efficient adapter transfer of self-supervised speech models for automatic speech recognition." ICASSP 2022-2022 IEEE International Conference on Acoustics, Speech and Signal Processing (ICASSP). IEEE, 2022.
[3] Otake, Shinta, Rei Kawakami, and Nakamasa Inoue. "Parameter efficient transfer learning for various speech processing tasks." ICASSP 2023-2023 IEEE International Conference on Acoustics, Speech and Signal Processing (ICASSP). IEEE, 2023.
[4] Peng, Junyi, et al. "Parameter-efficient transfer learning of pre-trained transformer models for speaker verification using adapters." ICASSP 2023-2023 IEEE International Conference on Acoustics, Speech and Signal Processing (ICASSP). IEEE, 2023.

I think a comparison with the standard adapter from [1] is required. To keep things as simple as possible, instead of the proposed TCAC, the adapter from [1] could potentially use the ECAPA-TDNN langauge/speaker embedding as a second input.

I’ll end with listing minor (formatting) issues:
* Line 20: inconsistent citation style
* Line 34: word ‘ControlNet’ missing
* Line 37: this challenge instead of these challenges (labeling data is a solution, not a challenge)
* Line 44,52: CA-SSLR instead of CA-SSL
* Line 76,78,80,84: hyperbolic language
* Line 84-86: This sentence does not make sense to me
* Line 87: inconsistent citation style
* Figure 1: “Condition-aware” instead of “conditioned-aware"
* Multiple instances where an abbreviation is explained while it has been used previously, e.g. CER in line 249,
* Figure 2 is unreadable in black and white print, using a different marker for each hue could alleviate this problem.
* Line 199: inconsistent citation style
* Line 280: requires instead of required
* Line 280: Real-Time Factors
* Most justifications in the paper checklist contain a spelling mistake, are missing, or do not refer correctly. Also, the instruction block was not deleted.

**Questions:**

I do not understand the time-channel attention conditioner. Given formula 1, what are \alpha, \gamma, and \beta? Are they a function? In line 164, they are defined as vectors? In line 166, you mention a linear layer to compute attention scores? How are these attention scores used? Do you not use a self-attention layer?

It is unclear to me how exactly the time-channel attention conditioner is used within the SSL model. Given formula 1, I assume the conditioner is inserted between each wav2vec 2.0 transformer layer?

In section 5.1, does the LID-FT experiment imply 2 fine-tunings, first on LID (with what data?), and then on ASR (with what data?), where the first fine-tuning updates the SSL model, and the second only the ASR decoder?

In section 5.1, what hyperparameters were used for these experiments? Is there a more comprehensive learning rate scan for the LID-TCAC setting compared to the XLSR-R baseline and LID-FT setting? Are they conducted with the same amount of update steps?

In section 5.1, line 236, it states “only TCAC layers are updated”, while I assume the ECAPA-TDNDN language ID decoder was also trained?

In section 5.2, Table 2 states ASR adapted, but aren’t the models adapted to language ID?

In section 5.2, how do you condition using the ground-truth language ID labels? In the same context, how are hard-predicted and soft-predicted language labels used for conditioning, and where do they come from?

In section 5.2, second paragraph, are the results shown in Figure 2 all your experiments? Regardless, how did you ensure a fair comparison (learning rate scan, number of updates)? Why did you choose to compare with LoRA? You mention that Chen et. al. [2023b] shows marginal improvements for LoRA, but this paper shows that AdapterBias and Houlsby are more effective for ASR?

**Limitations:**

While the proposed method improves performance for the specific condition of the ML-SUPERB benchmark, it might not work as well with unbalanced datasets (e.g., 100 hours of english data language, 1 hour of spanish and mandarin data).

---

> ### Author Rebuttal · Authors · 2024-08-07
>
> Thank you for your detailed comments, which have helped us improve our manuscript. Below are our clarifications:
>
> **Speaker Decoder and Generalization Ability:**
> We respectfully disagree that the speaker decoder detracts from the results. We intend to show that integrating conditioning in the SSL improves unseen tasks. In Table 1, CA-SSLR conditioners/LID-decoder are trained on LID loss, then frozen, and the ASR decoder is trained on top. Thus, no component in the CA-SSLR encoder is trained for ASR but obtains improvements w.r.t. Baseline. In the same manner, CA-SSLR$^L$ models improve SV by 20% while the CA-SSLR$^L$ encoder conditioners were adapted on ASR and LID losses, not SV. Therefore, the speaker decoder is essential for demonstrating the generalization ability of our condition-aware approach and showing that CA-SSLR$^L$ is a generalist encoder.
>
> **Comparison with Houlsby Adapter ([1], Chen et al. [2023b]):**
> CA-SSLR not only offers parameter-efficient fine-tuning (PEFT), but its principal contribution lies in incorporating dynamic adaptation to the language and speaker of the input, thereby improving generalizability, as noted by Reviewers iiaB and Jsg6. We performed additional experiments using the Houlsby adapter, as shown in Table A of the rebuttal PDF. Our ASR-CA-SSLR approach outperforms the ASR-Houlsby adapter, achieving WER of 18.6% and 31.6% compared to 20.3% and 34.6%, respectively. In SV, CA-SSLR improves EER from 1.29 to 1.15, while the Houlsby adapter worsens it from 1.29 to 1.37 compared to the XLS-R baseline. These results suggest that standard adaptation methods lack the enhanced generalization capabilities inherent to CA-SSLR. We appreciate the suggestions and plan to explore integrating our conditioner into the Houlsby adapter in future work.
>
> **Unbalanced Dataset and Performance**
> It's important to highlight that ML-SUPERB includes 20 few-shot languages with only five utterances each for ASR to evaluate performance on unbalanced datasets. Improvement in few-shot CERs is documented in Tables 1 to 3, Figure 2, Appendix Table 8, and additional Table A. These results demonstrate CA-SSLR's robustness in unbalanced scenarios.
>
> **Fair Comparison and Hyperparameter Search:**
> We confirm that the model parameters, learning rates, and batch size for ASR training are consistent with those in the ML-SUPERB benchmark paper. Parameters specific to the LID and SV decoders, such as dropout rates and hidden channels, were set using frozen SSLR settings to ensure a fair experimental setup. We added a reference in Section 4.2 to Appendix A.1 for clarification.
>
> **Enhancements in Presentation:**
> - Removed the duplicated content in Lines 121-131. This duplication was inadvertently introduced by one of the authors who didn’t notice that this content had been moved to another section.
> - Revised the hierarchical writing with Figure A.
> - Moved TCAC explanation before line 140 and substituted text with precise equations to assure reproducibility.
> - Consolidated SV and LID configurations in lines 141-146 into a standard setup, relocated to the experimental section.
> - Moved the information in 221-222 to the methods section.
> - Included suggested parameter-efficient transfer learning references [1~4] in the related work and fixed the minor formatting issues.
>
>
> We add the following equations for TCAC Implementation:
>
> The TCAC module process latent representations at layer $l$, $\mathbf{S}^{(l)}\in\mathbb{R}^{C\times T}$, and the latest estimate of the conditioning features $\mathbf{z}\in\mathbb{R}^R$ and generates modulated latent representations $\mathbf{\tilde{S}}^{(l)}$ as
>
> $$\mathbf{\tilde{S}} _{t,c}^{(l)} = \text{TCAC}(S _{t,c}^{(l)}, \mathbf{z}) = \tilde{\gamma} _{t,c}^{(l)}(\mathbf{z}, S^{(l)})S _{t,c}^{(l)}+ \tilde{\beta} _{t,c}^{(l)} (\mathbf{z}, S^{(l)})$$
>
> Thus, the latent features are modulated by time-channel dependent scales $\tilde{\gamma} _{t,c}^{(l)}$ and biases $\tilde{\beta} _{t,c}^{(l)}$, obtained by:
>
>
> $\tilde{\gamma}_{t,c}^{(l)} (\mathbf{z}, \mathbf{S}^{(l)})=\alpha_t^{(l)}(\mathbf{z}, \mathbf{S}^{(l)})\times \gamma_c^{(l)}(\mathbf{z})$
>
> $\tilde{\beta}_{t,c}^{(l)} (\mathbf{z}, \mathbf{S}^{(l)})=\alpha_t^{(l)} (\mathbf{z}, \mathbf{S}^{(l)}) \times \beta_c^{(l)}(\mathbf{z})$
>
> where channel-dependent $\gamma^{(l)},\beta^{(l)}\in \mathbb{R}^C$ are obtained as
>
> $\gamma^{(l)}(\mathbf{z})=\mathbf{W}_\gamma^{(l)}\mathbf{z}+\mathbf{b} _\gamma^{(l)}$
>
> $\beta^{(l)}(\mathbf{z}) = \mathbf{W}_\beta^{(l)} \mathbf{z} + \mathbf{b} _\beta^{(l)}$
>
> The time-dependent scales $\alpha^{(l)} \in \mathbb{R}^T$ are obtained with an additive attention mechanism as
>
> $\alpha^{(l)} _t(\mathbf{z}, \mathbf{S}^{(l)}) = \mathbf{v}^T _\alpha f(\mathbf{W} _\alpha^{(l)}  [\mathbf{S} _t^{(l)T} \mathbf{z}^T]^T$+ $\mathbf{b} _a^{(l)})$
>
> where $f(.)$ is a ReLU non-linearity, $\mathbf{W} _\alpha^{(l)}\in \mathbb{R}^{C'\times (C+R)}$, $\mathbf{b} _\alpha^{(l)}\in\mathbb{R}^{C'}$, and $\mathbf{v _\alpha}\in\mathbb{R}^{C'}$.
>
> We obtained the conditioning features $\mathbf{z}$ from the hard decisions or internal embedding layer $\mathbf{e}\in\mathbb{R}^E$ of the intermediate decoders, by $\mathbf{z} = \mathrm{LayerNorm}(\mathbf{W} \mathbf{e} + \mathbf{b})$, where the affine transform parameters $\mathbf{W}$, $\mathbf{b}$ are shared across TCAC layers.
>
> **Additional Clarifications:**
> - Alpha, beta, and gamma are functions that predict vectors, in parts of the text the dependency on (z,S) is dropped to keep the notation uncluttered.
> - The conditioning feature can be derived from the decoder's second last layer (embedding layer) or decoder output (either as hard or soft labels).
> - The LID decoder was updated with the pre-trained, frozen SSLR model but remained frozen during TCAC training, as shown in Figure A.
> - "ASR-adapted" refers to training CC/TCAC with ASR loss only.
> - TCAC is integrated within wav2vec 2.0 layers after the self-attention module. These details are included in the model architecture section.

---

> > ### Comment · Reviewer_xgMo · 2024-08-12
> >
> > Thanks for these clarifications and new experiments. I've raised my score, incorporating the new experiments comparing to Houlsby, and the inability to grok the updated presentation style fully (due to no fault of your own).

---

> > > ### Author Response · Authors · 2024-08-14
> > >
> > > Thank you for considering our clarifications and new experiments. We appreciate your adjusted score and will work to further improve the clarity of our presentation.

---

### Official Review · Reviewer_4eyk · 2024-07-13

**Soundness:** 4
**Presentation:** 3
**Contribution:** 3
**Rating:** 7
**Confidence:** 4

**Summary:**

This paper employs multi-task learning with hierarchical conditioning to adapt pre-trained speech SSL models. By utilizing lightweight task-related auxiliary decoders repeatedly at various positions, the method gradually tailors the SSL representations. A time-channel-dependent conditioner is introduced to facilitate the fusion process. By incorporating LID and Speaker SV decoders, the proposed approach achieves significant improvements in multi-lingual ASR, LID, and SV.

**Strengths:**

- The method is novel in reusing the decoders multiple times for incremental conditioning.
- The framework is general for various sets of speech tasks. One could change the conditioning decoder(s) according to the target tasks.
- Relatively strong results in adapting SSL models for multi-lingual ASR, which may inspire more SSL-based methods in multi-lingual scenarios.

**Weaknesses:**

Although the authors refer to the proposed model as "generalist", it still requires proper selection of conditioning/auxiliary decoders according to the target tasks. In the paper, the major task is multi-lingual ASR and the initial conditioning decoder is a language identifier, which are closely related. As illustrated in Table 4, further incorporating the SV conditioner leads to less performance gain.

A terming issue: the "Time-wise Attention" seems *time-dependent scaling* rather than an "attention mechanism". In Eq. (2), $\alpha_t^{(l)}(\textbf{z}, \textbf{S}^{(l)}) \in \mathbb{R}^T$ is computed frame-wise with a linear projection, and it scales the SSL features at different timesteps without interactions among frames/timesteps.

**Questions:**

(Line 144-145) It states "... the TCAC, SV, and LID decoders are trained...". But in Figure 1, the two decoders look frozen and the caption confirms this. Which is the actual setting?

(Line 163) It states that the output projection layers of conditioners are all shared. But as stated at line 221, each auxiliary decoder, at different points, takes the previous SSL representations using a weight average. Are those weights and input projection layers partially shared across positions, or totally position-dependent?

(Line 188) What augmentation is leveraged in the "Extended Few-shot condition"?

(Line 236) Which model does "the pre-trained and fixed SSLR model" refer to? If it refers to "XLS-R" in Table 1, then it is a bit confusing because a prediction head is added and the SSLR model is not purely pre-trained.

(Line 251) The conditioner takes $z \in \mathbb{R}^{C_E}$ as input, then how is it adpted to consume LID labels? Some illustrations here would avoid confusion.

(Table 4) Why is $\text{CA-SSLR}^{L,S}(\text{TCAC})$ not compared? As Eq. (3) mentions the adaptation method of this "full configuration", it is surprising that it does not appear in the final comparison.

Some personal suggestions/concerns regarding the presentation:

- Line 145: "ASR,a" -> "ASR,"
- Line 163 implies the re-estimation is conducted every three layers for both LID and SV, which conflicts with the actual setting.
- Table 4: the results are derived by *jointly optimization* for all three tasks, including system "**+ FT**", but this operation is not explicitly mentioned in the experiment setting, which could be confusing at first.

**Limitations:**

The authors have adequately addressed the limitations and potential negative societal impact of their work.

---

> ### Author Rebuttal · Authors · 2024-08-07
>
> Thanks for the comment. Here are some clarifications.
>
> Regarding the **generalist model concern**, we want to emphasize that CA-SSLR is considered a generalist model because it maintains the base model's integrity while improving performance on previously unseen tasks.  In Table 1, CA-SSLR conditioners/LID-decoder are trained on LID loss, then frozen, and the ASR decoder is trained on top. Thus, no component in the CA-SSLR encoder is trained for ASR but obtains improvements w.r.t. Baseline. In the same manner, CA-SSLR$^L$ models improve SV by 20% while the CA-SSLR$^L$ encoder conditioners were adapted on ASR and LID losses, not SV, showing that CA-SSLR$^L$ is a generalist encoder.
>
> We acknowledge the concern about using **the term "Time-wise Attention."** Although it seems like time-dependent scaling, it fits the concept of additive attention, without normalizing weights to sum up to one with softmax. We observed that removing the softmax provided better results. Our approach involves using a linear projection to compute $\alpha _t^{(l)}(z, S^{(l)} _{t,c}) \in \mathbb{R}^T$  by concatenating a single feature vector $\mathbf{z}$ with $S^{(l)} _{t,c}$ at each timestep, similar to the method used in [A]. This process assigns different weights to each timestep, allowing us to scale $S^{(l)} _{t,c}$ accordingly. Even though there are no direct interactions between timesteps, the dynamic assignment of weights based on features aligns with the principles of attention, as the approach applied in [B].
>
> In Table 4, we have updated **CA-SSLR$^{L,S}$ TCAC results** for comparison with XLS-R, achieving the best results in LID and ASR and ranking second in SV, showing strong adaptation capabilities while maintaining competitive generalization ability.
>
> | Model                                    | LID (10min) | ASR | EER | CDF  | LID (1h) | ASR | EER | CDF
> |------------------------------------------|----------|----------|----------|----------|----------|----------|----------|----------|
> | XLS-R                                    | 89.0     | 29.0     | 1.29     | 0.093    | 90.9     | 22.7     | 1.29     | 0.093    |
> | + CA-SSLR$^{L,S}$ (CC)             | **89.1** | 18.8     | **1.04** | **0.075**| 88.1     | 15.0     | **0.94** | **0.073**|
> | + CA-SSLR$^{L,S}$(TCAC)           | 89.0     | **18.3** | 1.11     | 0.086    | **93.5** | **14.4** | 1.01     | 0.077    |
>
> Regarding the **confusion about frozen and trainable parameters**, first, we train the LID decoder on the pre-trained SSL model. The decoder gets a weighted average of the SSL encoder layers. Afterward, this LID decoder is then frozen and used to get the conditioning features that are fed to the LID TCAC conditioners. In this case the LID decoder gets a weighted average of the CA-SSLR encoder layers computed up to that point, and the conditioning feature is recomputed every three CA-SSLR layers. Here, we only train the linear projection before the LID decoder and the LID TCAC parameters to the LID-conditioned CA-SSLR encoder, termed CA-SSLR$^L$. Similarly, the SV decoder is trained on top of the CA-SSLR$^L$ then frozen. Then, we just train the SV TCAC parameters to get the CA-SSLR$^{L,S}$. The adaptation training has been detailed in **Figure A of the rebuttal PDF** and will be added to Sec. 3. We also update lines 144 and 145 to avoid confusion.
>
> Regarding **parameter sharing**, the only layer-dependent features used for calculating the condition feature z are the linear projections for the decoders and the weights used for the weighted sum of the SSL layers before the linear predictions, as shown in Figure A of the rebuttal PDF. All other decoder parameters are shared.
>
> For the **extended few-shot condition**, the original ML-SUPERB settings include "normal languages" with 10 minutes to 1 hour of data per language and "few-shot languages" with only 5 utterances each. In the "extended few-shot condition," we incorporate the language labels from these few-shot data for LID training but continue using only 5 utterances with transcriptions for ASR training. This is because language labels are more accessible to obtain than transcriptions. We will add this table to the appendix to clarify these settings.
>
> | **Data Per Language**           | **Language Type**    | **LID Training**                      | **ASR Training**                         |
> |-------------------------|----------------------|---------------------------------------|------------------------------------------|
> | Original Settings       | Normal    |  10 min - 1 hr      | 10 min - 1 hr  with transcription          |
> |                         | Few-shot  |  Not used for LID                   | 5 utts with transcriptions     |
> | Extended Few-shot       | Extended Few-shot   | 10 min - 1 hr          | 5 utts  with transcriptions     |
>
>
> Regarding whether the **"pre-trained and fixed SSLR model" refers to XLS-R**, Yes, it does refer to the frozen pre-trained XLS-R model with a trained decoder head. We will update the sentence to "frozen XLS-R model with ASR decoder" for clarity.
>
> Regarding the **condition feature z**, we have updated the sentence in L163 for clarity: "We re-estimate the condition feature 𝑧 from the updated language or speaker embedding every three layers using a linear projection and layer normalization, which are shared across layers.” This has been revised to “As shown in Figure A, 𝑧 is derived from the updated language or speaker embedding through a linear projection layer or an embedding layer for hard LID/SV labels.”
>
>
> [A] Luong, M. T., Pham, H., & Manning, C. D. (2015, September). Effective Approaches to Attention-based Neural Machine Translation. In Proceedings of the 2015 Conference on Empirical Methods in Natural Language Processing (pp. 1412-1421).
>
> [B]Lin, Z., Feng, M., dos Santos, C. N., Yu, M., Xiang, B., Zhou, B., & Bengio, Y. (2022, July). A STRUCTURED SELF-ATTENTIVE SENTENCE EMBEDDING. In International Conference on Learning Representations.

---

> > ### Comment · Area_Chair_5pJo · 2024-08-12
> >
> > Dear Reviewer 4eyk,
> >
> > As the discussion between the reviewers and authors is coming to an end, could you please respond to the authors to confirm whether your concerns have been addressed?
> >
> > Thanks!
> >
> > AC

---

### Official Review · Reviewer_mDeV · 2024-07-13

**Soundness:** 3
**Presentation:** 3
**Contribution:** 3
**Rating:** 6
**Confidence:** 4

**Summary:**

This paper introduced a framework, CA-SSLR, that integrates conditioning into pre-trained Self-Supervised Learning (SSL) models by adapting only the trainable conditioner. Through a hierarchical self-conditioning mechanism, CA-SSLR can match or achieve better the performance of single-task fully fine-tuned models and benefit from more efficient model tuning. CA-SSLR offers a versatile and efficient approach to integrating conditioning information into pre-trained models.

**Strengths:**

The strengths of the paper are:

1. Good presentation of the proposed framework.
2. Strong results on different tasks and different scenarios.

**Weaknesses:**

The weaknesses of the papers are that the overall system might be complicated because of performing multiple tasks and some tasks would be dependent on the output of other tasks. As a result, the inference cost would be increased for the subsequent tasks.

**Questions:**

Here are some minor concerns to the paper:
1. Is it necessary to provide the rel RTF in Table 3 and 4? I feel like the absolute value would be enough though.

2. It is intuitive that the multilingual ASR model can be improved by using LID as the condition. Is there a cascaded baseline doing LID + ASR that the pretrained model is not fused? The results can help us understand the real effectiveness of the proposed framework.

3. Since the proposed framework achieves better parameter efficient finetuning, is there a training speed or peak memory usage comparison  between the proposed method and full finetuning? How about its efficiency compared to other PEFT methods, like LoRA?

**Limitations:**

The advantage of the paper is to use less resources than full finetuning but achieve similar performance. However, the inference cost is increased. This can be problematic when the method is used in real applications. The inference cost might weigh more than the training cost when the training cost is affordable. Another limitation is that the framework cannot be used for streaming purposes as mentioned by the authors.

---

> ### Author Rebuttal · Authors · 2024-08-07
>
> Thank you for the insightful comment. Here are some clarifications.
>
> Regarding **inference cost**, the encoder parameters are shared among all three tasks, which helps to minimize expenses. The LID decoder is lightweight, with an RTF of less than 0.001, so it doesn't significantly add to the computational load of the ASR task. While the ASR and SSL models are more complex, with single-pass RTFs of 0.004 and 0.016, respectively, they may need multiple inferences based on the decoding algorithm. As shown in Tables 9 and 10, the CA-SSLR system achieves 30 to 45% faster RTF than multi-task baselines across different components and scenarios.
>
> Concerning the **cascaded baseline**, we examined the cascading baseline for LID + ASR in Table 2 and Figure 2. In this method, we train the LID decoder with a pre-trained SSL model and use another SSL model with TCAC components to train the ASR decoder. Our findings indicate that the TCAC can achieve performance comparable to a fully fine-tuned approach for languages with abundant resources and surpasses the fully fine-tuned approach in few-shot ASR languages.
>
> To assess **training speed and peak memory usage**, we compared the CA-SSLR approach with the additional baseline, Houlsby Adapter, and a fully fine-tuning approach (the results comparing the Houlsby Adapter and CA-SSLR$^{L}$ are detailed in Table A of the rebuttal PDF). We evaluate the training speech for 10k iterations with batch size 8, and the results are shown in the following table:
>
>
> | **Method**     | **Bottleneck Dims.**             | **Training Speed**                          | **Peak Memory Usage** |
> |---------------|------|---------------------------------------------|-----------------------|
> | Houslby Adapter | 256  | 76 mins          |            58B           |
> | CA-SSLR$^{L}$(3L) | 256 | 120 mins          |             68B          |
> | FT              | -  | 135 mins                            |         79B             |
>
> we found that the CA-SSLR approach ranks second compared to the Houlsby Adapter and a fully fine-tuning approach in speed and memory usage. However, CA-SSLR surpasses the Houlsby Adapter in adaptation effectiveness, as shown in Table A(b) of the rebuttal PDF. It also demonstrates **the best generalization ability among the three methods, detailed in Tables 1, 4, and A**. Additionally, we acknowledge that the current implementation is not yet optimal and are committed to further improvements.

---

> > ### Comment · Reviewer_mDeV · 2024-08-12
> > **Reply to the rebuttal**
> >
> > Thanks for the rebuttal. My concerns have been addressed. If the explanations can be included in the revised version, I am willing to raise my score to 6.

---

> > > ### Author Response · Authors · 2024-08-14
> > >
> > > Thank you for your feedback. We’re glad to hear that your concerns have been addressed. We will definitely include the explanations in the revised version.

---

### Official Review · Reviewer_iiaB · 2024-07-14

**Soundness:** 3
**Presentation:** 3
**Contribution:** 3
**Rating:** 7
**Confidence:** 5

**Summary:**

The paper introduces a method named CA-SSLR, a versatile model for various speech-processing tasks that integrates language and speaker embeddings from earlier layers to reduce reliance on input audio features while preserving the base model's integrity. More specifically, both LID and SID conditioning features are integrated as additional inputs for the SSL model. The proposed approach shows advantages in improving the downstream speech tasks.  ML-SUPERB dataset is used in the evaluation, which shows superior performance over the conventional SSL models.

**Strengths:**

- The paper is well-written and easy to follow. The topic is interesting, and the proposed idea is simple yet effective.
- The experiments are thorough and convincing.

**Weaknesses:**

- There is no comparison between the proposed approach and the SOTA numbers on ML-SUPERB. Readers would likely be interested in understanding the gap between the proposed approach and the SOTA numbers.
- It is somewhat unclear when to use the proposed TCAC, as sometimes using CC (without attention) seems sufficient, as shown in Table 2 and Table 4. More analysis should be provided to explain these results, rather than just mentioning them.

**Questions:**

1. The first question relates to the second point in the weaknesses: when is attention necessary? Can we visualize the attention using one or two examples?
2. Which layers are helpful in generating the embeddings? 3L and 4L are reported in the tables; are the shallow layers sufficient?

**Limitations:**

See the Weaknesses.

---

> ### Author Rebuttal · Authors · 2024-08-07
>
> Thank you for the helpful comments. Here are some clarifications.
>
> First, for **the generalist model without fine-tuning, the SOTA results** for pre-trained SSLR models can be found in the paper ML-SUPERB challenge [A], where the MMS-1b model performs the best with 1hr ASR WER 18.1% and LID accuracy 86.1% respectively (see the table below). However, we used XLS-R 300M to make our experiments feasible.
>
> Second, **with fine-tuning, the single-task SOTA results** can be found in Table 1 and Figure 2, with 90.1% LID accuracy and 17.3% ASR CERs.
>
>
>
> | **SSL Model**    | Model size            | **10mins LID**          |  **10mins ASR**                |       **10mins ASR**           | **1hr LID**               |      **1hr ASR**            |    **1hr ASR**              |
> |-------------|-----------------|------------------------|-----------------|-----------------|----------------------|-----------------|-----------------|
> |               |                | Normal      | Normal | Few-shots   | Normal      | Normal | Few-shots   |
> | XLS-R~\citep{shi2023ml} | 0.3B     | 66.9                   | 29.2            | 40.9            | 87.9                 | 22.0            | 39.3            |
> | MMS-1b ~\citep{shi2023findings} | 1B | 84.8               | 21.3            | 30.2            | 86.1                 | 18.1            | 30.8            |
> | XLS-R (Ours)   | 0.3B              | 89.0                   | 29.0            | 39.0               | 90.9                 | 22.7            | 36.9               |
> | + Embed TCAC^{L}  | 0.3B            | 89.0                   | 17.8            | 31.8            | 90.9                 | 13.5            | 31.4            |
> | + CA-SSLR^{L} (TCAC, 3L) | 0.3B | 88.6                | 18.6            | 31.6            | 93.4                 | 15.1            | 29.6            |
>
>
> We found that compared with the MMS-1b baseline, the proposed system achieves better results in Normal languages and comparable results in few-shots languages than the model with three times more parameters (300M vs. 1B). We will add these results in Table 3.
>
> Regarding the **concern about using shallow layers**, we would like to clarify that the CA-SSLR system recomputes the embeddings every 3 or 4 layers rather than predicting only once in the initial layers, as shown in Table A in the rebuttal PDF. From the SSL feature weights utilized by the LID and SV decoders, we observe that LID weights are evenly distributed, while SV weights are more concentrated in the earlier layers. Consequently, incorporating predictions from higher LID layers can enhance LID accuracy and improve conditioning features. Conversely, for SV, using only the first 6 or 12 layers of XLS-R might be sufficient. We will further explore this in future work.
>
> While CC applies the same transformation to all time frames, we added the attention module in **TCAC to be able to emphasize different time-frames** of the encoded features within the SSLR layers.  When deciding between TCAC and CC, our experiments indicate that TCAC provides better results for LID and ASR. However, as shown in Table 4, CC currently yields slightly better results for SV. Overall, relative improvements of TCAC in ASR and LID are larger than relative degradation in SV. We plan to conduct further investigations to enhance the SV performance of TCAC. In the updated version, we will provide visualizations of the attention mechanism for the TCAC module using one or two examples. We will further provide a more detailed analysis of our results in the revised version.
>
> [A] Shi, J., Chen, W., Berrebbi, D., Wang, H. H., Huang, W. P., Hu, E. P., ... & Watanabe, S. (2023, December). Findings of the 2023 ml-superb challenge: Pre-training and evaluation over more languages and beyond. In 2023 IEEE Automatic Speech Recognition and Understanding Workshop (ASRU) (pp. 1-8). IEEE.

---

> > ### Comment · Reviewer_iiaB · 2024-08-11
> >
> > Thanks for the answers. I'm suggesting putting the previous SOTA results (with some descriptions) in the paper as a reference for the readers. I'll keep my rating of the paper.

---

> > > ### Author Response · Authors · 2024-08-11
> > >
> > > Thank you for your positive feedback and suggestion. We will add the SOTA results along with descriptions to the paper.

---

### Official Review · Reviewer_Jsg6 · 2024-07-21

**Soundness:** 3
**Presentation:** 2
**Contribution:** 3
**Rating:** 6
**Confidence:** 3

**Summary:**

This paper introduces conditioning into self-supervised learning of speech representations. In particular, a hierarchical self-conditioning mechanism is introduced where intermediate language and speaker embeddings are used to condition upper layers. The proposed approach is used together with XSLR and mHubert models and several experiments are conducted using the SUPERB benchmark.

**Strengths:**

- Multiple experiments are conducted demonstrating the benefits of the proposed method.

- A novel conditioning method is introduced for learning speech representations in a self-supervised way.

**Weaknesses:**

I think the main weakness of the paper is that it is too dense. Too much information is presented which makes several sections of the paper hard to follow. For example, it's not very easy to follow section 3. It would help if additional equations are added or a more detailed figure is included. Fig. 1 is too high-level and it's not very informative.

In addition, some sentences from sections 1 and 2 are literally copied and pasted in section 3. For example, the following sentence can be found both in sections 2 and 3 "Unfortunately, this results in employing a distinct encoder per task, leading to a large increase in computational load that scales linearly with the number of tasks to be assessed.". It would good if the authors rephrase such sentences. There are a few more such examples.

**Questions:**

Please see above.

**Limitations:**

Limitations have been addressed.

---

> ### Author Rebuttal · Authors · 2024-08-07
>
> Thanks for the insightful comments. We have included additional equations and figures to improve the clarity and address the writing in Sec. 3.
>
> Regarding additional figures, we add **Figure A in the rebuttal PDF** to clarify our system as the elaboration for the original Figure 1.
>
>
> Regarding additional equations, we further add **details equations to clarify the Time-Channel Attention Conditioner to form a separate subsection of Sec 3.2** to improve the readability:
>
> "As depicted in Fig.1b, the TCAC module ingests the latent representations of the CA-SSLR at layer $l$, $\mathbf{S}^{(l)}\in\mathbb{R}^{C\times T}$, and the latest estimate of the conditioning features $\mathbf{z}\in\mathbb{R}^R$ and generates modulated latent representations $\mathbf{\tilde{S}}^{(l)}$ as
>
> $$\mathbf{\tilde{S}} _{t,c}^{(l)} = \text{TCAC}(S _{t,c}^{(l)}, \mathbf{z}) = \tilde{\gamma} _{t,c}^{(l)}(\mathbf{z}, S^{(l)})S _{t,c}^{(l)}+ \tilde{\beta} _{t,c}^{(l)} (\mathbf{z}, S^{(l)})$$
>
> where $t$, $c$, and $l$ represent time, channel, and layer indices, respectively. Thus, the latent features are modulated by time-channel dependent scales $\tilde{\gamma} _{t,c}^{(l)}$ and biases $\tilde{\beta} _{t,c}^{(l)}$. These are obtained as the products:
>
> $\tilde{\gamma} _{t,c}^{(l)} (\mathbf{z}, \mathbf{S}^{(l)})=\alpha _t^{(l)}(\mathbf{z}, \mathbf{S}^{(l)})\times  \gamma _c^{(l)}(\mathbf{z})$
>
> $\tilde{\beta}_{t,c}^{(l)} (\mathbf{z}, \mathbf{S}^{(l)})=\alpha_t^{(l)} (\mathbf{z}, \mathbf{S}^{(l)}) \times \beta_c^{(l)}(\mathbf{z})$
>
> where channel-dependent $\gamma^{(l)},\beta^{(l)}\in \mathbb{R}^C$ are obtained as
>
> $\gamma^{(l)}(\mathbf{z})=\mathbf{W}_\gamma^{(l)}\mathbf{z}+\mathbf{b} _\gamma^{(l)}$
>
> $\beta^{(l)}(\mathbf{z}) = \mathbf{W}_\beta^{(l)} \mathbf{z} + \mathbf{b} _\beta^{(l)}$
>
> The time-dependent scales $\alpha^{(l)} \in \mathbb{R}^T$ are obtained with an additive attention mechanism as
>
> $\alpha^{(l)} _t(\mathbf{z}, \mathbf{S}^{(l)}) = \mathbf{v}^T _\alpha f(\mathbf{W} _\alpha^{(l)}  [\mathbf{S} _t^{(l)T} \mathbf{z}^T]^T$+ $\mathbf{b} _a^{(l)})$
>
> where $f(.)$ is a ReLU non-linearity, $\mathbf{W} _\alpha^{(l)}\in \mathbb{R}^{C'\times (C+R)}$, $\mathbf{b} _\alpha^{(l)}\in\mathbb{R}^{C'}$, and $\mathbf{v _\alpha}\in\mathbb{R}^{C'}$.
>
> We obtained the conditioning features $\mathbf{z}$ from the hard decisions or internal layer (embedding layer) $\mathbf{e}\in\mathbb{R}^E$ of the intermediate decoders, by $\mathbf{z} = \mathrm{LayerNorm}(\mathbf{W} \mathbf{e} + \mathbf{b})$, where the affine transform parameters $\mathbf{W}$, $\mathbf{b}$ are shared across TCAC layers.
>
> Thus, TCAC enables the model to dynamically adjust its behavior to the input audio in response to the provided conditioning features."
>
> Also, for the **Condition-Aware Learning Mechanism**, we remove Lines 121-131 to eliminate duplication and focus the section on our unique contribution. This duplication was inadvertently introduced by one of the authors who didn’t notice that this content had been moved to another section. Following are the revisions of the second and third paragraphs in Sec. 3.1:
>
> “The proposed CA-SSLR model improves the efficiency of evaluating multiple tasks by introducing a hierarchy of conditioners within the pre-trained SSL encoder layers. It utilizes intermediate predictions from the language identification (LID) and speaker verification (SV) decoders as conditions to recursively adapt subsequent layers, as shown in Figure A. This hierarchical approach structures the SSL encoder layers so that each layer refines its output based on the predictions of the preceding layers. Early layers produce embeddings that capture the essential language and speaker characteristics, which are then used to inform scaling and bias adjustments in later layers.
>
> We propose a novel mechanism, the Time-Channel Attention Conditioner (TCAC), which modulates the encoder's hidden representations using time-channel-dependent scales and biases. This approach enables the SSL encoder to dynamically adjust to varying tasks and input conditions. The inputs to these conditioners are embeddings derived from intermediate evaluations of the LID and SV decoders."

---

> > ### Comment · Area_Chair_5pJo · 2024-08-12
> >
> > Dear Reviewer Jsg6,
> >
> > As the discussion between the reviewers and authors is coming to an end, could you please respond to the authors to confirm whether your concerns have been addressed?
> >
> > Thanks!
> >
> > AC

---

> > ### Comment · Reviewer_Jsg6 · 2024-08-13
> >
> > I have read the rebuttal and based on it and I have increased my score from 5 to 6.

---

> > > ### Author Response · Authors · 2024-08-14
> > >
> > > Thank you for reviewing our rebuttal and for increasing your score. We appreciate your feedback on the writing and will continue to refine it to ensure clarity and quality.

---

### Author Rebuttal · Authors · 2024-08-07

Dear Reviewers,

We thank the reviewers for their insightful and positive feedback! We are encouraged that they appreciate various aspects of CA-SSLR, including the novelty (Reviewers Jsg6, 4eyk), the clarity and presentation of our writing (Reviewers iiaB, mDev), and the impressive experimental results (Reviewers Jsg6, iiaB, mDev, 4eyk) that demonstrate the benefits and general applicability of our CA-SSLR across multiple tasks and scenarios.

Your time and effort dedicated to improving our work are truly appreciated. We have answered all your questions and addressed the issues in detail in our rebuttal and the latest revision.

These revisions include additional explanations, paragraphs, and equations to help readers understand the proposed method and additional experiments to highlight its advantages.

Most importantly, we have added new results from the Houlsby adapter experiments further to illustrate CA-SSLR's impact on adaptation and generalization abilities (Reviewer xgMo), as well as results for the MMS-1b SSL model to update the current SOTA baseline (Reviewer iiaB). This response offers a high-level overview of these revisions for the convenience of reviewers and future readers.

Major revisions include:

- New Experiment Results in Sec. 5.2 and Sec. 5.4: A new experimental study of the **Houlsby adapter model** (Reviewer xgMo).
- New Experiment Results in Sec. 5.3 and Sec. 5.4: A **new SOTA baseline** of the MMS-1b SSL model and CA-SSLRL,S$^{L,S}$(TCAC) (Reviewers iiaB, 4eyk).
- Improved Writing in Sec. 3: Added **system figure and equations for TCAC** and removed duplicate sentences (Reviewers Jsg6, xgMo).

Minor revisions include:
- A detailed table of the extended few-shot settings is included in the appendix (Reviewer 4eyk).
- An analysis of training speed and peak memory usage was added in the appendix (Reviewer mDeV).


We hope these revisions address your concerns and improve the overall quality of our paper.

Thank you again for your review!

Best regards,

Authors

---

### Decision · Program_Chairs · 2024-09-25

**Decision:**

Accept (poster)

**Comment:**

The reviewers commend the paper for conducting multiple experiments that effectively demonstrate the benefits of the proposed method. They highlight the introduction of a novel conditioning technique for learning speech representations in a self-supervised manner. The paper is praised for being well-written, easy to follow, and presenting an interesting and effective idea. The experiments are described as thorough and convincing, with a well-presented framework. The paper shows relatively strong results in adapting self-supervised learning (SSL) models for multi-lingual automatic speech recognition (ASR), which could inspire more SSL-based methods in multi-lingual scenarios.